# Hyperglycaemia induces metabolic dysfunction and glycogen accumulation in pancreatic β-cells

Melissa F. Brereton[1], Maria Rohm[1], Kenju Shimomura[1], Christian Holland[1], Sharona Tornovsky-Babeay[2], Daniela Dadon[3], Michaela Iberl[1], Margarita V. Chibalina[4], Sheena Lee[1], Benjamin Glaser[2], Yuval Dor[3], Patrik Rorsman[4], Anne Clark[4] & Frances M. Ashcroft[1]

Insulin secretion from pancreatic β-cells is impaired in all forms of diabetes. The resultant hyperglycaemia has deleterious effects on many tissues, including β-cells. Here we show that chronic hyperglycaemia impairs glucose metabolism and alters expression of metabolic genes in pancreatic islets. In a mouse model of human neonatal diabetes, hyperglycaemia results in marked glycogen accumulation, and increased apoptosis in β-cells. Sulphonylurea therapy rapidly normalizes blood glucose levels, dissipates glycogen stores, increases autophagy and restores β-cell metabolism. Insulin therapy has the same effect but with slower kinetics. Similar changes are observed in mice expressing an activating glucokinase mutation, in *in vitro* models of hyperglycaemia, and in islets from type-2 diabetic patients. Altered β-cell metabolism may underlie both the progressive impairment of insulin secretion and reduced β-cell mass in diabetes.

[1] Department of Physiology, Anatomy and Genetics and OXION, University of Oxford, Parks Road, Oxford OX1 3PT, UK. [2] Endocrinology and Metabolism Service, Hadassah-Hebrew University Medical Center, Jerusalem 91120, Israel. [3] Department of Developmental Biology and Cancer Research, The Institute for Medical Research Israel-Canada, The Hebrew University-Hadassah Medical School, Jerusalem 91120, Israel. [4] Oxford Centre for Diabetes, Endocrinology and Metabolism, University of Oxford, Churchill Hospital, Oxford OX3 7LJ, UK. Correspondence and requests for materials should be addressed to F.A. (email: frances.ashcroft@dpag.ox.ac.uk).

The hallmark of the pancreatic β-cell is its ability to respond to glucose with increased insulin secretion. This process is impaired in diabetes, leading to chronic elevation of the blood glucose concentration. Long-term hyperglycaemia has deleterious effects in many tissues. In β-cells, it causes a reduction in insulin release, in insulin granule density and in β-cell number, a phenomenon termed glucotoxicity[1,2]. Numerous studies have examined the effects of hyperglycaemia on β-cell structure and function, both *in vitro*, using isolated islets or insulin-secreting cell lines, or *in vivo*, using obese diabetic animal models, but few have examined the time dependence and reversibility of the effects of hyperglycaemia, or the mechanisms involved. We have therefore investigated the progressive changes in β-cell dysfunction produced by diabetes, and their reversal, using an inducible mouse model of neonatal diabetes caused by an activating mutation in the ATP-sensitive potassium (K$_{ATP}$) channel[3,4].

The K$_{ATP}$ channel couples blood glucose levels to insulin secretion by virtue of its sensitivity to changes in β-cell metabolism. Elevation of blood glucose stimulates glucose uptake and metabolism by the β-cell, thereby increasing intracellular ATP. This closes K$_{ATP}$ channels and leads to β-cell depolarization, calcium influx and insulin granule exocytosis[5]. Gain-of-function mutations in either the Kir6.2 (*KCNJ11*) or SUR1 (*ABCC8*) subunit of the K$_{ATP}$ channel are a major cause of neonatal diabetes, a rare inherited disorder characterized by the development of diabetes within the first 6 months of life[6,7]. In neonatal diabetes, failure of the K$_{ATP}$ channel to close in response to metabolically generated ATP prevents β-cell electrical stimulation and insulin release, causing hypoinsulinaemia and hyperglycaemia[6–8]. Sulphonylurea drugs are used to treat patients with neonatal diabetes as they selectively block the open K$_{ATP}$ channels[9]. This results in far better glycaemic control, with fewer fluctuations in plasma glucose concentration, a lower HbA1C, a reduced incidence of hypoglycaemia and a simpler medication regime[10–15].

Recent studies demonstrate that chronic hyperglycaemia in mice expressing an activating K$_{ATP}$ channel mutation results in striking β-cell remodelling[3,16]. This is characterized by a marked loss of insulin granules, a reduction in islet number and dysregulation of α- and β-cell identity genes[3,16].

We now show that diabetes causes marked changes in β-cell carbohydrate metabolism and glucose-stimulated ATP production as soon as 24 h after exposure to hyperglycaemia. Prolonged diabetes also resulted in substantial glycogen accumulation that led to increased β-cell apoptosis. Restoring euglycaemia induced autophagy, rapidly dissipated glycogen stores, normalized metabolism and reversed many of the gene expression changes. Analysis of mice expressing an activating mutation in glucokinase revealed that enhanced glucose metabolism, rather than glucose *per se*, was responsible for the changes in gene expression and glycogen storage. Glycogen accumulation was observed also in islets of patients with type-2-diabetes. The progressive impairment of β-cell function with chronic hyperglycaemia helps explain why sulfonylurea therapy is more effective in neonatal diabetes patients with shorter diabetes duration[10], and has implications not only for the aetiology and treatment of neonatal diabetes, but also for other forms of diabetes.

## Results

**K$_{ATP}$ channel activation results in rapid diabetes**. We investigated the time course of the β-cell changes induced by hyperglycaemia and its reversal *in vivo* in an inducible mouse model of neonatal diabetes (βV59M)[3]. Nutrient-stimulated insulin secretion was switched off in βV59M mice at 12–14 weeks of age by β-cell-specific expression of an activating K$_{ATP}$ channel mutation (Kir6.2-V59M) commonly found in human neonatal diabetes[3,7]. This resulted in blood glucose levels >28 mM within 2 days. Euglycaemia could be restored by subcutaneous administration of the sulphonylurea glibenclamide, which closes the open K$_{ATP}$ channels, or by insulin[3].

No differences in plasma lipid levels were found between control mice and diabetic βV59M mice (Supplementary Fig.1). Free fatty acids, total serum cholesterol, HDL cholesterol, LDL/VHDL cholesterol were unchanged. Triglycerides were slightly but not significantly elevated. Aminoalanine transferase (ALT) activity, a marker of liver damage, was also unaffected. Thus the changes we observe are a result of hyperglycaemia/hypoinsulinaemia and not a secondary consequence of altered lipid metabolism.

**Diabetes duration impacts β-cell function**. Diabetes was associated with progressive changes in β-cell mass and ultra-structure. β-cell mass, assessed as the percentage of insulin staining per cm$^2$ of pancreas, was markedly lower in islets from 2- or 4-week diabetic βV59M mice (Fig. 1a). Islet density also fell, reflecting a decrease in both islet number and size (Fig. 1b). The reduction in insulin-labelled cells was paralleled by an increase in glucagon-positive cells (Fig. 1c). There was also a time-dependent decrease in insulin granule density, as shown by electron microscopy (EM), and a gradual development of large areas of unstructured cytoplasm in β-cells (Fig. 1d) that increased with the duration of diabetes (Fig. 1e). Hyperglycaemia for 24 h, however, had no effect on islet insulin labelling, granule number or islet ultrastructure (Fig. 1c,d).

Blood glucose levels in diabetic βV59M mice were rapidly normalized with either insulin[3] or sulphonylurea (glibenclamide) therapy (Fig. 2a). However, the ability of sulphonylureas to restore euglycaemia was dependent on diabetes duration, as seen in human patients with neonatal diabetes[10]. Following 2 weeks of diabetes, glibenclamide normalized blood glucose within 24 h in 88% (7/8) of mice, but it was only successful in 47% (7/15) of mice after 4 weeks of diabetes. Only mice in which euglycaemia was restored within 48 h were used in this study (Fig. 2a). Islet insulin content was unaffected at 1 week of diabetes but decreased by 50% after 2 weeks of diabetes (Fig. 2b): in our earlier studies, it fell by 70% after 4 weeks of diabetes[3]. Glibenclamide therapy both prevented, and reversed, changes in islet insulin content (Fig. 2b).

Both glibenclamide and insulin treatment also reversed the ultrastructural and histochemical changes induced by diabetes but the time course was very different (Fig. 2c,d). Remarkably, in mice exposed to 4 weeks of diabetes, β-cell ultrastructure was completely normalized within 24 h of starting glibenclamide therapy. By contrast, substantial areas of electrolucent cytoplasm remained after 24 h of euglycaemia in insulin-treated mice (Fig. 2d,e): however, these were largely absent after 1 week of euglycaemia (Fig. 2d).

**Functional consequences of altered glucose handling in β-cells**. We next explored the functional consequences of progressive diabetes on β-cell function by measuring glucose-stimulated changes in NAD(P)H autofluorescence and ATP content in freshly isolated islets from control and diabetic βV59M mice.

In control islets, increasing glucose caused a clear dose-dependent increase in both NAD(P)H autofluorescence (Fig. 3a,b) and ATP content (Fig. 3d). By contrast, glucose elevation elicited much smaller increases in NAD(P)H fluorescence in islets from 24-h diabetic βV59M mice, and minimal change after 4 weeks of diabetes (Fig. 3a,b). Similarly, glucose did not elicit an increase in ATP content in 4-week diabetic islets (Fig. 3d). There was no significant difference in

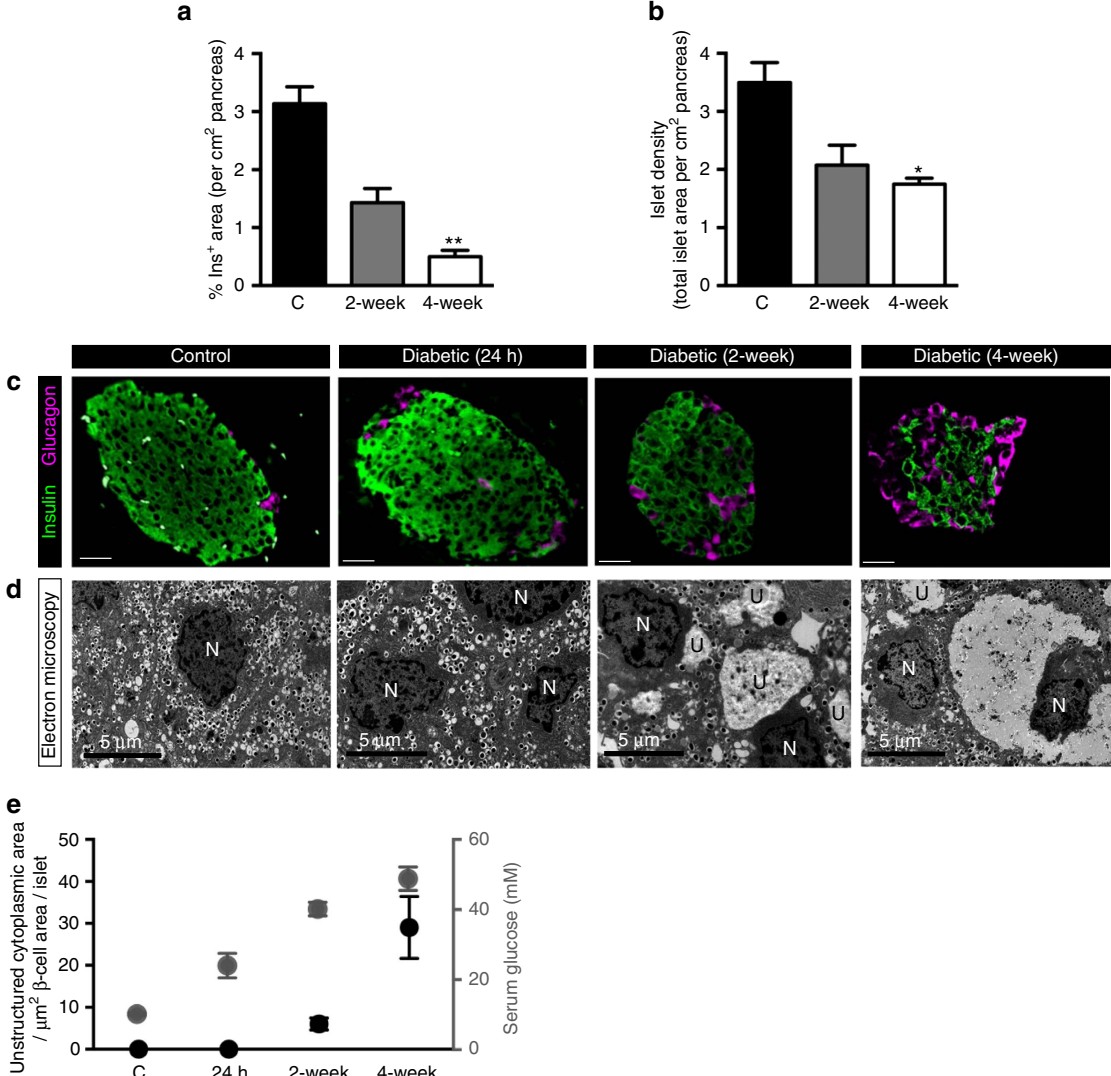

**Figure 1 | Hyperglycaemia in βV59M mice induces progressive changes in β-cell mass and ultrastructure.** (**a**,**b**) Mean islet cross-sectional area immunostaining for insulin (**a**), and total islet area (**b**), expressed as a percentage of the total cross-sectional area of the pancreas ($cm^2$) in control mice (black bar; $n = 8$), and βV59M mice left diabetic for 2 weeks (gray bar; $n = 5$) or 4 weeks (white bar; $n = 4$). Data are mean ± s.e.m. (5 sections per mouse, 100 µm apart). *$P < 0.05$; compared with control; one-way ANOVA followed by *post hoc* Bonferroni test. (**c**,**d**) Representative pancreatic sections from control mice (column 1), βV59M mice left diabetic for 24 h (column 2), 2 weeks (column 3) and 4 weeks (column 4). (**c**) Islets were immunostained for insulin (green) and glucagon (pink). Scale bars 200 µm. (**d**) Electron microscopy. N, nucleus. U, unstructured substance. Scale bars 5 µm. Data are representative of 3-4 mice in each case. (**e**) Serum glucose measurements (white circles) and area of unstructured cytoplasm in β-cells (black circles, calculated from electron micrographs) in control, 24-h, 2-week and 4-week diabetic βV59M mice. For serum glucose measurements 'n' corresponds to number of mice where, $n = 26$ (control), $n = 6$ (24-h diabetic), $n = 6$ (2-week diabetic) and $n = 8$ (4-week diabetic). For calculation of unstructured cytoplasm 'n' corresponds to the number of islets from 3–8 mice where, $n = 8$ (control), $n = 3$ (24-h diabetic), $n = 4$ (2-week diabetic) and $n = 4$ (4-week diabetic). Data are mean ± s.e.m.

ATP concentration at basal glucose (2 mM) between control and diabetic islets.

Remarkably, these effects were rapidly reversed by *ex vivo* culture at low glucose. Culture of 4-week βV59M diabetic islets at 5 mM glucose for 48 h partially restored both the NAD(P)H (Fig. 3c) and ATP (Fig. 3e) responses to 20 mM glucose. Addition of the sulphonylurea gliclazide (which closes $K_{ATP}$ channels) produced an even greater effect. Culture of 4-week βV59M diabetic islets at 25 mM glucose, however, did not restore the ATP response (Fig. 3e), indicating that reversal is a consequence of low glucose rather than *ex vivo* culture.

The changes in β-cell metabolism induced by diabetes were accompanied by striking changes in expression of metabolic genes.

Microarray analysis of mRNA extracted from isolated islets demonstrated that expression of numerous genes was significantly altered in response to 2 weeks of hyperglycaemia. Over 780 genes were upregulated > 1.5-fold, and > 300 genes were downregulated: full details are given at (http://www.ebi.ac.uk/arrayexpress/ for experiment E-MTAB-5086). Importantly, many genes involved in carbohydrate metabolism were also strongly affected (Supplementary Table 1a,b).

To validate these changes in key metabolic genes, we performed quantitative PCR using islets from control mice, βV59M mice exposed to diabetes for 24 h or 4 weeks, and βV59M mice exposed to 4 weeks of diabetes followed by 4 weeks of glibenclamide therapy. We focused on genes involved in

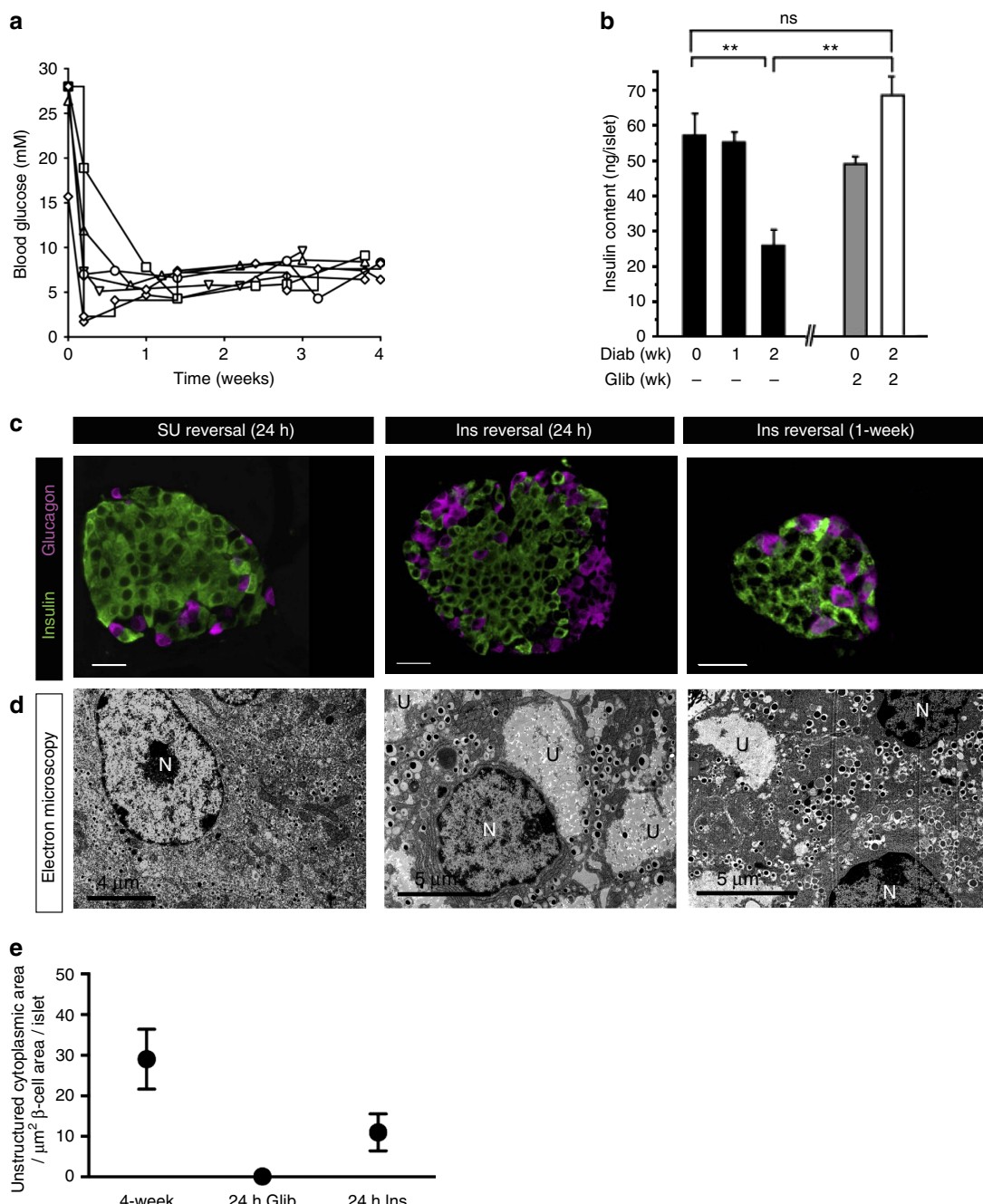

**Figure 2 | Effects of hyperglycaemia reversal in βV59M mice.** (**a**) Blood glucose levels of 4-week diabetic βV59M mice implanted with a slow release sulphonylurea pellet at time zero for mice in which euglycaemia was established within 4 days. (**b**) Mean (±s.e.m.) insulin content of isolated islets ($n = 4$–12 mice). Mice were implanted with either placebo or glibenclamide pellets immediately after gene induction and elevation of blood glucose to >20 mM. The duration of diabetes is indicated above and that of glibenclamide therapy below. *$P < 0.05$; *, $P < 0.01$. NS, not significant ($t$-test). (**c,d**) Representative pancreatic sections from 4-week diabetic βV59M mice after 24-h treatment with the sulphonylurea glibenclamide (SU reversal, left), or treatment with insulin for 24 h (middle) or 1 week (right). (**c**) Islets immunostained for insulin (green) and glucagon (pink). Scale bar (200 μm). (**d**) Electron microscopy. N, nucleus. U, unstructured substance. Scale bars 4 μm (left), 5 μm (middle, right). Data are representative of 3–4 mice in each case. (**e**) Mean (±s.e.m.) area of unstructured cytoplasm in β-cells (calculated from electron micrographs of isolated islets) from 4-week diabetic βV59M mice ($n = 4$ islets) and 4-week diabetic βV59M mice treated for 24 h with glibenclamide ($n = 4$ islets) or insulin ($n = 6$ islets).

gluconeogenesis (Fig. 3f), glycolysis (Fig. 3g), the Krebs cycle (Fig. 3h), and mitochondrial metabolism (Fig. 3i) that were substantially altered in the microarray data.

It is evident that some genes changed their expression within 24 h of diabetes, whereas others did not. For example, *G6pc2*, *Pfkfb2*, *Ogdh1* and *Cox6a2* were significantly affected by 4 weeks of diabetes but were unchanged by 24 h of hyperglycaemia.

Conversely, *Pdk1*, *Fbp1*, *Fbp2* and *Ppp1r3c* were fully upregulated within 24 h: thus, these genes may be under different metabolic control than those that take longer to change. Some of these gene changes were reversed by 4 weeks of glibenclamide therapy (for example, *G6pc2*, *Pfkfb2*, *Ogdh1*, *Cox6a2* and *Ppp1r3c*) but, interestingly, many others were not (for example, *Pdk1*, *Fbp1*, *Fbp2*).

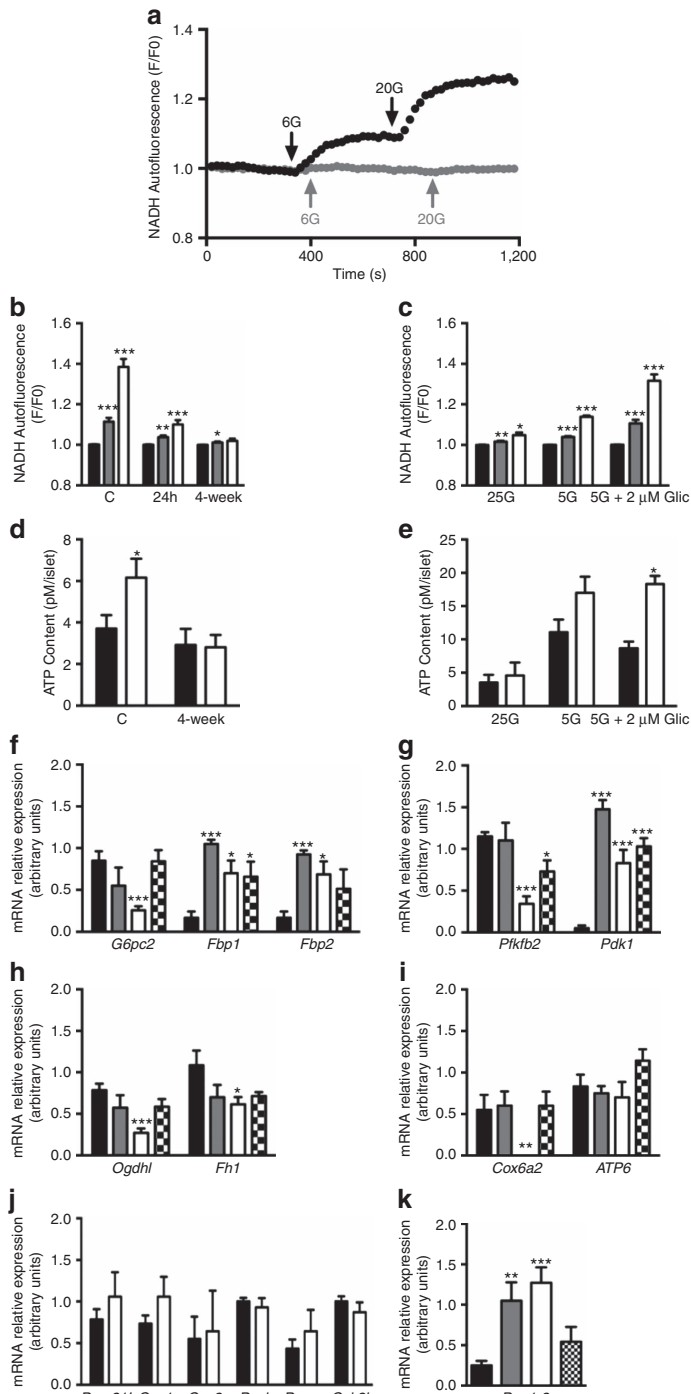

**Figure 3 | Diabetic βV59M mice have altered glucose metabolism and gene expression.** (**a**) Representative changes in NAD(P)H autofluorescence measured in response to 6 mM and 20 mM glucose from islets isolated from a control mouse (black trace) or a 4-week diabetic βV59M mouse (gray trace). Data are normalized to the level in 1 mM glucose. (**b**) Mean (± s.e.m.) NAD(P)H autofluorescence produced by 1 mM (black bars) 6 mM (gray bars) and 20 mM glucose (white bars) in control islets (n = 7 islets from 3 mice) and islets isolated from 24 h (n = 7 islets from 4 mice) and 4-week diabetic βV59M mice (n = 9 islets from 4 mice). *P < 0.05, ** P < 0.01, ***P < 0.001 v. 2 mM glucose (t-test). (**c**) Mean (± s.e.m.) NAD(P)H autofluorescence produced by 1 mM (black bars) 6 mM (gray bars) and 20 mM glucose (white bars) in islets isolated from 4-week diabetic βV59M mice and then cultured at 25 mM glucose (n = 5 islets), 5 mM glucose (n = 8 islets) or 5 mM glucose plus 2 μM gliclazide (n = 8 islets) for 72 h. *P < 0.05, ** P < 0.01, ***P < 0.001 v. 2 mM glucose (t-test). (**d**) Mean (± s.e.m.) ATP content of control (C; n = 6 mice) and 4-week diabetic βV59M (4-week n = 5 mice) islets incubated for 1 h in 2 mM (black bars) or 20 mM glucose (white bars). *P < 0.05 versus control (t-test). (**e**) Mean (± s.e.m.) ATP content of islets incubated at 2 mM glucose (black bars) or 20 mM glucose (white bars) for 1 h. Islets were isolated from 4-week diabetic βV59M mice (n = 4 mice) and cultured at 25 mM glucose, 5 mM glucose, or 5 mM glucose plus 2 μM gliclazide, or 25 mM glucose for 72 h before experiment. *P < 0.05 versus control (t-test). (**f–k**) Quantitative PCR of mRNA isolated from islets from control mice (black bars; n = 6 mice) or from βV59M mice 24 h (gray bars; n = 4 mice) and 4 weeks (white bars; n = 7 mice) after diabetes onset, or after 4 weeks of diabetes and 4 weeks of glibenclamide therapy (hatched bars; n = 7 mice). Genes assayed are involved in gluconeogenesis (**f**) glycolysis (**g**), Krebs cycle (**h**), oxidative phosphorylation (**i**), and glycogen metabolism (**j,k**). Data are mean ± s.e.m. *P < 0.05, ** P < 0.01, ***P < 0.001 v. control (t-test).

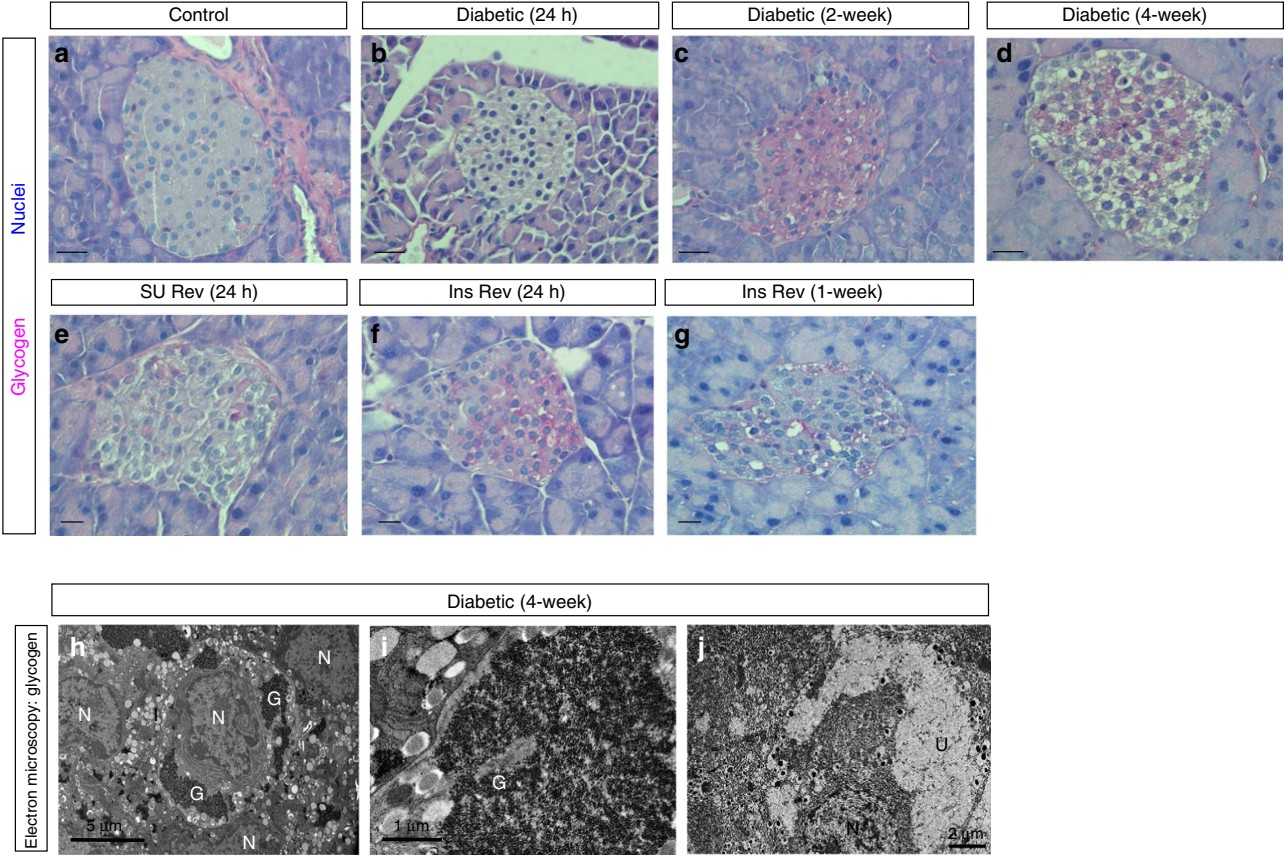

**Figure 4 | Glycogen accumulates in βV59M mice. (a–g)** Representative pancreatic sections showing PAS staining for glycogen (pink) from control mice (**a**); βV59M mice left diabetic for 24 h (**b**), 2 weeks (**c**) or 4 weeks (**d**); and βV59M mice left diabetic for 4 weeks and then treated with glibenclamide (SU) for 24 h (**e**) or with insulin (Ins) for 24 h (**f**) or 1-week (**g**). Scale bars, 200 μm. (**h–j**) Electron micrographs from 4-week diabetic βV59M mice obtained using lead-citrate fixation (**h,i**), which preserves glycogen granules, or using conventional fixation (**j**), which leads to loss of glycogen and produces an electrolucent cytoplasm. Note insulin granules appear pale and without a 'halo' using lead-citrate fixation. N, nucleus; G, glycogen; U, unstructured substance ( = glycogen); I, insulin granules. Scale bars 5 μm (left), 1 μm (middle) 2 μm (right).

Overall, the gene changes we observed are expected to reduce flux through the Krebs cycle and oxidative phosphorylation, and increase gluconeogenesis. For example, downregulation of the glycolytic enzyme 6-phosphofructo-2-kinase/fructose-2,6-bisphosphatase (*Pfkfb2*) and upregulation of pyruvate dehydrogenase kinase (*Pdk1*) would reduce entry of acetyl CoA into the Krebs cycle (Fig. 3f,g). In addition, downregulation of the Krebs cycle enzymes fumarase (*Fh1*) and oxoglutarate dehydrogenase (*Ogdh1*) would decrease entry of reducing equivalents (NADH) into the electron transport chain (Fig. 3h). This, together with the reduced expression of cytochrome c oxidase (*Cox6a2*) mRNA, the last enzyme in the electron transport chain, explains why glucose fails to stimulate NAD(P)H and ATP production in diabetic βV59M islets (Fig. 3i). However, ATP synthase (*ATP6*) did not change.

Upregulation of the gluconeogenic genes *Fbp1* and *Fbp2* (which catalyse hydrolysis of fructose 1,6-bisphosphate to fructose-6-phosphate) and downregulation of *G6pc2* (which catalyses the hydrolysis of glucose-6-phosphate to glucose) (Fig. 3f), implies that glucose entering the β-cell will accumulate as glucose-6-phosphate (G6P). In turn, this will lead to glycogen storage, as G6P is a potent allosteric stimulator of glycogen synthase[17] and glucose is an inhibitor of glycogen phosphorylase. Most enzymes directly involved in glycogen synthesis did not change their expression (Fig. 3j), but *Ppp1r3c* (PTG), which activates glycogen synthase, was strongly upregulated after 24 h (Fig. 3k).

**Glycogen accumulation in diabetic β-cells.** Collectively, these gene changes, together with the enhanced substrate availability and the impaired ATP production, suggest that glucose entering diabetic βV59M β-cells will be stored as glycogen. This was investigated by both light microscopy (using Periodic Acid Schiff (PAS) staining) and electron microscopy.

No PAS staining was detected in control β-cells (Fig. 4a) or β-cells exposed to hyperglycaemia for 24 h (Fig. 4b). However, considerable glycogen staining was evident in β-cells exposed to high glucose for >2 weeks (Fig. 4c). After 4 weeks of diabetes, many β-cells appeared 'clear' or 'empty' (Fig. 4d), due to loss of glycogen during processing[18]. Such loss becomes more evident when glycogen deposits are very substantial and leads to an apparent decrease in glycogen staining and the appearance of an 'empty' cytoplasm. Strikingly, no glycogen was detected in 4-week diabetic βV59M β-cells after 24 h of glibenclamide therapy (Fig. 4e). By contrast, substantial glycogen staining remained in insulin-treated mice after 24 h of euglycaemia (Fig. 4f), and this was not fully abolished even after 1 week of euglycaemia (Fig. 4g).

Ultrastructural studies using lead-citrate fixation to preserve glycogen revealed extensive deposits of glycogen particles in most β-cells of βV59M islets after 4 weeks of hyperglycaemia (Fig. 4h,i). Indeed, in some β-cells the glycogen content was so excessive that it constituted the majority of the cytoplasm and caused distortion of the nucleus and endoplasmic reticulum. However, no glycogen was found in control islets. It is important to note that glycogen is

not detected using conventional EM fixatives, as it is water soluble and easily washed out of cells during fixation. This leaves large regions of electrolucent 'unstructured' cytoplasm, like those shown in Fig. 4j (and Figs 1d and 2d, which correspond to the 'empty' cells shown in Fig. 4d).

**Metabolic dysfunction disrupts protein turnover.** Glycogen exists as a large macromolecular complex with numerous associated proteins, and its accumulation in other tissues is controlled by autophagy as well as metabolism[19,20]. We therefore next explored whether autophagy is impaired in diabetic βV59M islets.

Proteins marked for autophagic degradation are first ubiquitinated and then targeted to the autophagosome via p62, and both ubiquitin and p62 commonly accumulate in cells in which autophagy is inhibited[21–25]. Several lines of evidence indicate that autophagic flux is impaired in 4-week diabetic βV59M islets. First, there was a marked increase in ubiquitin protein aggregates, at both light microscope and ultrastructural levels (Fig. 5a-c). Secondly, p62 protein was also increased (Fig. 5d,e). Interestingly, ubiquitin and p62 colocalized with glycogen deposits in 4-week diabetic βV59M β-cells (Fig. 5c,d, Supplementary Fig. 2A,B). Thirdly, LC3B-II protein levels were enhanced (Fig. 5e; Supplementary Fig. 2A,B). Finally, transcript levels of beclin-1, which is known to promote autophagy[26], were significantly reduced (Fig. 5f).

Autophagy was very low in control β-cells and no change in the number of autophagosomes could be detected in 4-week diabetic βV59M β-cells (Fig. 5g). However, restoration of normoglycaemia led to a dramatic increase in the number of autophagosomes (Fig. 5g,h). Glycogen granules were found in both autophagosomes (Fig. 5i) and lysosomes (Fig. 5l), which suggests that when diabetes is reversed, glycogen is degraded, at least in part, via the autophagic–lysosomal system.

**Hyperglycaemia induces metabolic dysfunction in T2DM.** We next explored whether the metabolic changes we observed in βV59M islets are a general feature of hyperglycaemia. Glycogen accumulation was observed in human islets from donors with type-2 diabetes at both light (Fig. 6a) and electron (Fig. 6b) microscopy levels. β-Cells with large areas of 'empty' cytoplasm, corresponding to regions where glycogen was washed away during processing, were also observed in islets from type-2 diabetic donors (* marked areas in Fig. 6a). Glycogen also accumulated in β-cells following *in vitro* culture of islets from non-diabetic human donors (Fig. 6c-e) or control mice (Supplementary Fig. 3) for 48 h at 25 mM glucose. Accumulation of glycogen by control mouse islets cultured at high glucose has been reported previously[27–29]. The NAD(P)H response to glucose elevation was also impaired in mouse islets cultured at 25 mM glucose (Supplementary Fig. 3). As observed for diabetic βV59M β-cells, glycogen stores were rapidly dissipated when islets were returned to 5 mM glucose for 24 h (Fig. 6e; Supplementary Fig. 3).

Similar results were found for INS-1 cells: culture at high glucose caused glycogen accumulation, which was reversed when cells were subsequently cultured at low glucose (Fig. 6f–h; Supplementary Fig. 4). In INS-1 cells treated with the lysosomal inhibitors E-64d or pepstatin A, glycogen was found in defined puncta corresponding to LAMP2 positive lysosomes (Supplementary Fig. 5). The gene expression changes found in diabetic βV59M islets were also largely mirrored in INS-1 cells treated with high glucose for 24 h. However, in contrast to 4-week diabetic βV59M islets, almost all gene expression changes were reversed by 24-h culture at 2 mM glucose (Fig. 7i–m).

To determine if altered glucose metabolism underlies glycogen formation, we used a mouse model that expresses an activating mutation in the glucokinase gene (GCK) specifically in adult β-cells (mutGCK mice)[30]. Glucokinase catalyses the phosphorylation of glucose to glucose-6-phosphate, the rate-limiting step in glucose metabolism and mutGCK mice display hypermetabolism and enhanced insulin secretion despite normal blood glucose levels[30] Within 4 days of transgene induction, we found substantial accumulation of glycogen in β-cells by both light (Fig. 6n) and electron (Fig. 6o) microscopy. Furthermore, RNAseq analysis of islets isolated from mutGCK mice showed similar gene changes to those found in βV59M islets and INS-1 cells cultured at high glucose concentrations. This included elevation of *Ppp1r3c* (3.4-fold), *Pdk1* (2.3-fold), *Fbp2* (4-fold), and *Aldob* (68-fold), and a decrease in *G6pc2* (2.4-fold), *Cox6a2* (1.7-fold), and *Pfkfb2* (1.5-fold). This suggests that glucose metabolism, rather than glucose *per se*, drives both glycogen formation and the metabolic gene expression changes observed in diabetic β-cells.

**Impaired metabolism and glycogen storage underlie β-cell death.** It is well established that diabetes or impaired autophagy leads to apoptosis in β-cells[31,32]. Marked β-cell apoptosis is also observed in mutGCK mice following prolonged activation of the transgene[30]. We therefore explored whether altered metabolism and glycogen accumulation contribute to β-cell death in diabetes.

In a previous study[3], we found no obvious increase in the number of apoptotic nuclei after 4 weeks of diabetes when assessed by nuclear morphology (chromatin condensation) with electron microscopy. However, this approach may underestimate the extent of apoptosis as few cells can be examined, apoptosis and cell debris removal are quickly complete, and earlier time points were not assessed. Thus, we examined earlier steps in the β-cell death pathway, which are more readily detectable. Figure 7a,b shows that cleaved caspase-3 was not present in control islets but significant labelling was detected in βV59M islets after 2 weeks of hyperglycaemia, by which time glycogen accumulation was extensive. Furthermore, active caspase-3 was found with glycogen in insulin-positive cells (arrow), but was absent in glycogen-negative cells (arrowhead) and in glucagon-positive α-cells (which did not accumulate glycogen (Fig. 7d). Apoptotic bodies ('cellular blebbing'), a hallmark of apoptosis were also seen in electron micrographs of β-cells of 4-week diabetic βV59M islets (Fig. 7e,f). Interestingly, these apoptotic bodies were filled with glycogen granules when measures were taken to preserve glycogen (Fig. 7f).

To further examine the relationship between glycogen accumulation and β-cell death, we manipulated glycogen levels independently of the ambient glucose level. First, we used the antidiabetic drug metformin, which inhibits glycogen synthesis[33], to reduce glycogen levels at high external glucose concentrations. In the presence of 25 mM glucose, metformin prevented glycogen accumulation in INS-1 cells (Fig. 7g,h) and reduced caspase-3 activation and apoptosis (Fig. 7i). Furthermore, expression of *Ppp1r3c* (PTG) was reduced (Fig. 7j). This protein targets protein phosphatase-1 (PP1) to glycogen and its expression is closely linked to glycogen content[34]. In cells in which glycogen had already accumulated, subsequent addition of metformin did not affect glycogen or cleaved caspase-3 levels (Supplementary Fig. 6). This suggests metformin, at least at the concentration we used, does not itself reduce caspase-3 activation.

Secondly, we further enhanced glycogen levels at high glucose concentrations by silencing UDP-glucose-pyrophosphorylase-2 (UGP2) in INS-1 cells. A significant decrease in both UGP2 mRNA (Fig. 7k) and protein (Fig. 7l; Supplementary Fig. 7A) was

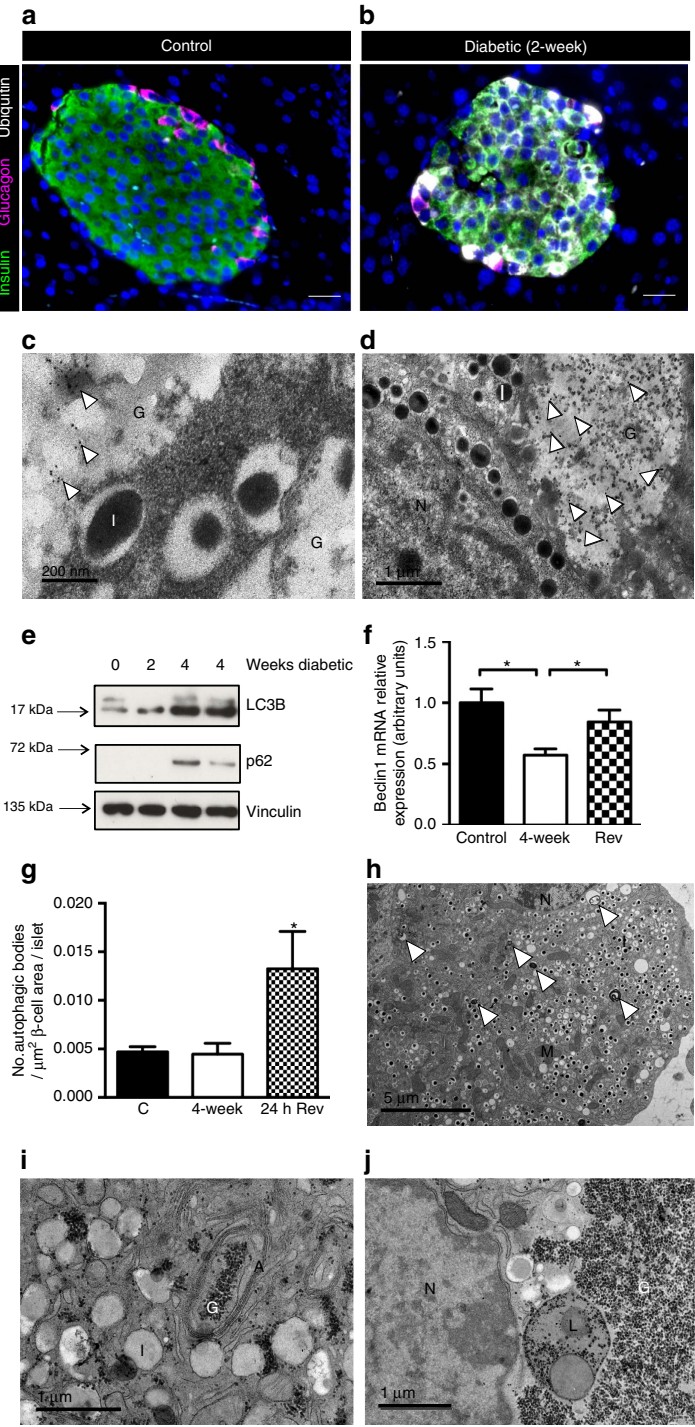

**Figure 5 | βV59M mice have altered protein turnover.** (**a**,**b**) Representative pancreatic sections immunostained for insulin (green), glucagon (pink) and ubiquitin (white) from control mice (**a**) and 4-week diabetic βV59M mice (**b**). Scale bars 200 μm. (**c**,**d**) Representative electron micrographs of islets isolated from 4-week diabetic βV59M mice showing immunogold labelling for ubiquitin (**c**, arrowed) or p62 (**d**, arrowed). Scale bars 200 nm (**c**), 1 μm (**d**). I, insulin granule. G, glycogen. N, nucleus. (**e**) Western blot analysis for LC3B, p62 and vinculin (loading control) in islets isolated from control, 2-week and 4-week diabetic βV59M mice. The lower band in the LC3B blot is LC3B-II. Data are representative of 3 experiments. (**f**) Quantitative PCR of *beclin-1* mRNA in islets isolated from control mice (black bars; $n = 6$ mice) and βV59M mice 4 weeks after diabetes onset (white bars; $n = 7$ mice), or after 4 weeks of diabetes and 4 weeks of glibenclamide therapy (hatched bars; $n = 7$ mice). Data are mean ± s.e.m., $n = 6$-8 mice per genotype. *$P < 0.05$ (*t*-test). (**g**) Mean ( ± s.e.m.) density of autophagosomes in islets of control mice (black bars, $n = 8$ mice) and βV59M mice after 4 weeks of diabetes (white bars, $n = 4$ mice), or after 4 weeks diabetes plus 24-h diabetes reversal with glibenclamide (hatched bars, $n = 4$ mice). *$P < 0.05$ versus control (c, *t*-test). Autophagosomes were counted from 10 fields of view per islet and expressed per μm² of β-cell cytoplasm. (**h–j**) Representative electron micrographs from 4-week diabetic βV59M mouse treated with glibenclamide for 48 h (24-h euglycaemia). Arrowheads (**h**) denote autophagosomes. A, autophagosomes. G, glycogen granules. N, nucleus. I, insulin granule. L, lysosome. M, mitochondrion. Scale bar 5 μm (**h**), 1 μm (**i**,**j**).

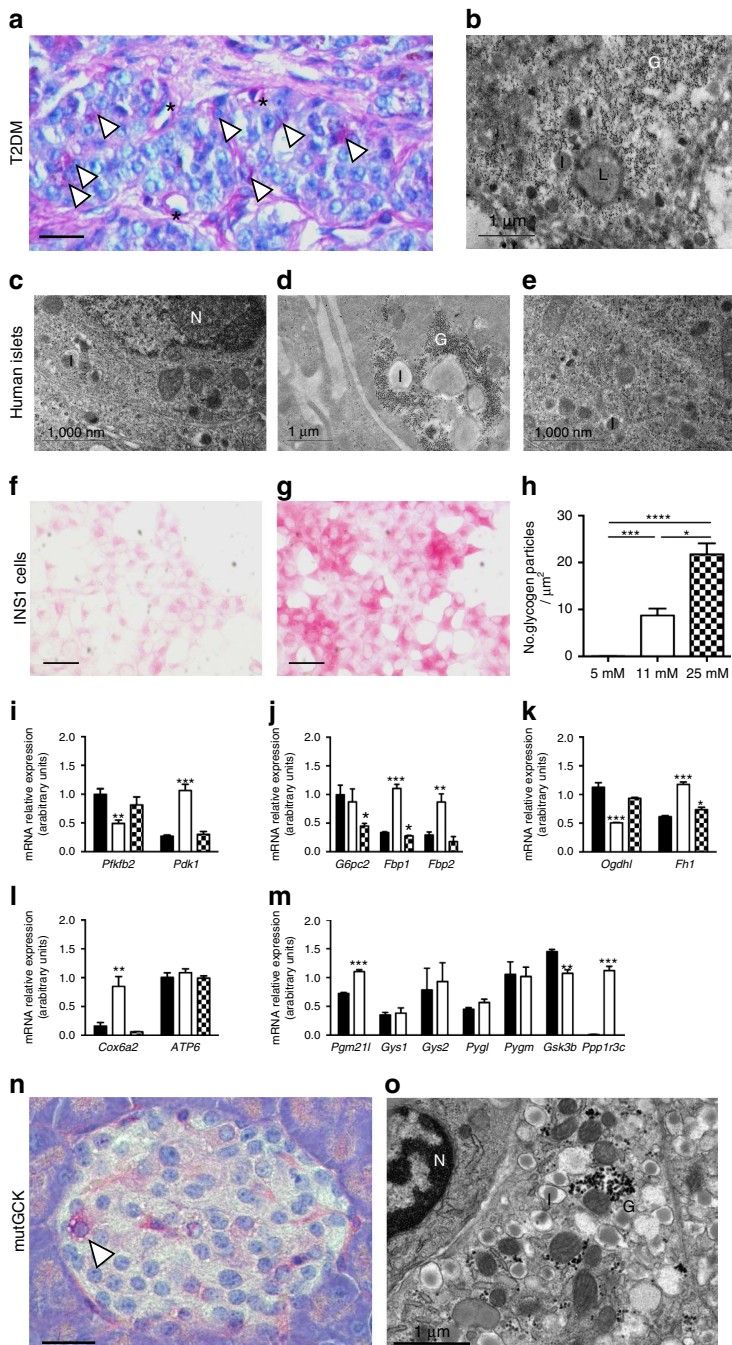

**Figure 6 | Hyperglycaemia alters glucose handling in human and rodent islets, and in INS-1 cells. (a,b)** Representative light microscope pancreatic section (**a**) and β-cell electron micrograph (**b**) from two different organ donors with type 2 diabetes. In each case, data are representative of 12 patients with type 2 diabetes and fasting blood glucose levels of 10–16 mM. (**a**) Section stained for glycogen with PAS (pink) and for nuclei with haematoxylin (blue). Scale bar, 200 μm. Arrowheads denote glycogen; * denotes an 'empty' cell. (**b**) G, glycogen. I, insulin granule. L, lysosome. Scale bar 1 μm. (**c–e**) Representative electron micrographs of isolated islets from non-diabetic organ donors following 48-h culture at 5 mM (**c**) or 25 mM glucose (**d**), or 48 h at 25 mM glucose followed by 24 h in 5 mM glucose (**e**). Data are representative of islets from three donors. I, insulin granule. N, nucleus. G, glycogen granules. Scale bars 1 μm. (**f,g**) Representative images of PAS-stained INS-1 cells following 48-h culture in 5 mM (**f**) or 25 mM glucose (**g**). Scale bars 200 μm. (**h**) Mean ( ± s.e.m.) number of glycogen particles measured from electron micrographs in INS-1 cells cultured for 48 h at 5 mM glucose ($n = 29$ cells), 11 mM glucose ($n = 37$ cells), and 25 mM glucose ($n = 24$ cells). *$P < 0.05$, ** $P < 0.01$, ***$P < 0.001$ (Mann–Whitney test). (**i–m**) Quantitative PCR of mRNA isolated from INS-1 cells cultured for 24 h at 2 mM (black bars) or 25 mM glucose (white bars); or 24 h at 25 mM glucose followed by 24 h at 2 mM glucose (hatched bars). Genes assayed are involved in gluconeogenesis (**i**), glycolysis (**j**), the Krebs cycle (**k**), oxidative phosphorylation (**l**), and glycogen metabolism (**m**). Data are mean ± s.e.m. ($n = 4$). *$P < 0.05$, ** $P < 0.01$, ***$P < 0.001$ versus 2 mM glucose (t-test). (**n,o**) Representative pancreatic section stained for glycogen (PAS; pink) and nuclei (haematoxylin; blue) (**n**) and β-cell electron micrograph (**o**) from mutGCK mice 4 days after gene induction. (**n**) Arrow indicates a β-cell containing glycogen. Scale bar, 200 μm. (**o**) N, nucleus. I insulin granule. G, glycogen particles. Scale bar, 1 μm.

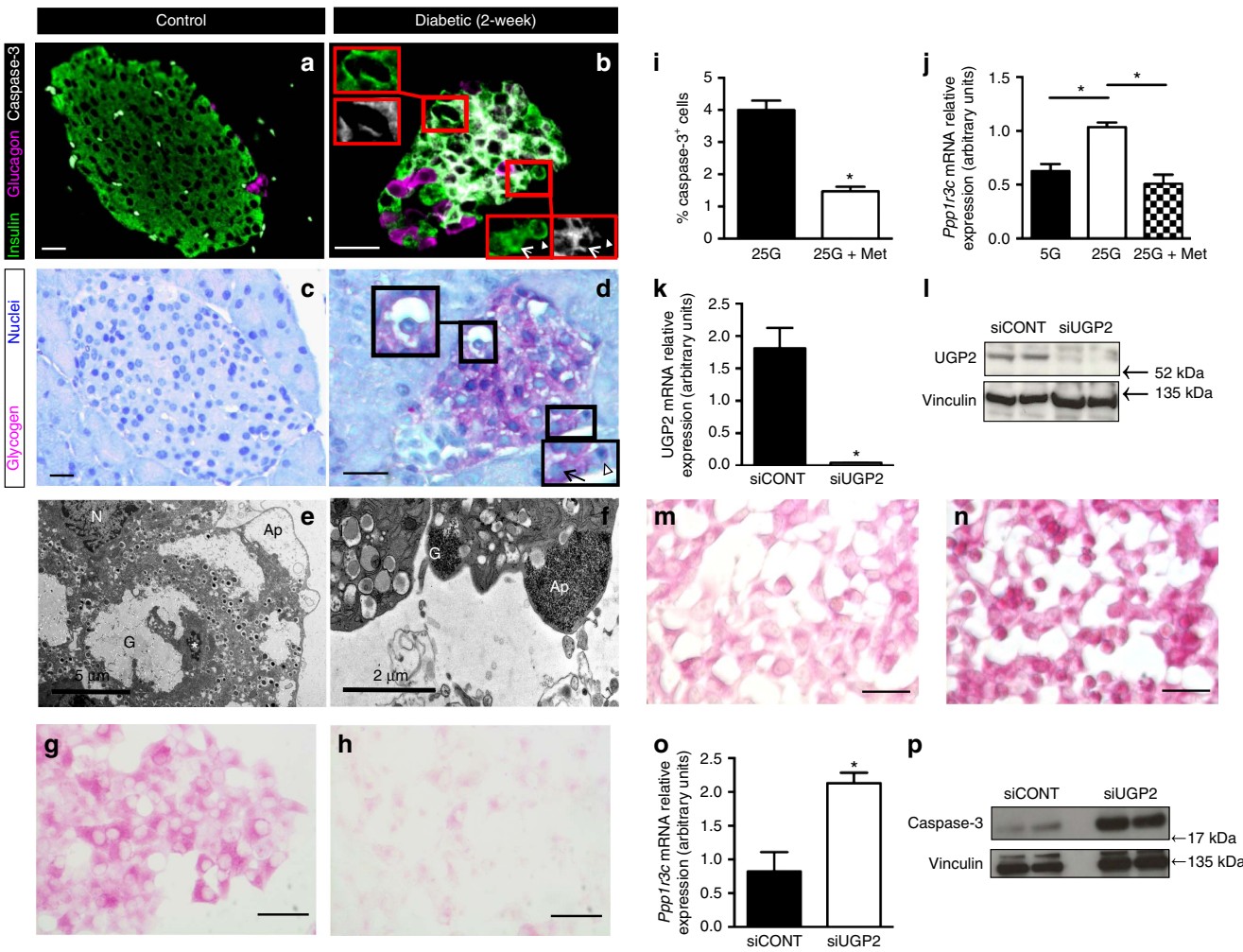

**Figure 7 | Hyperglycaemia in βV59M mice promotes apoptosis. (a–d)** Representative serial pancreatic sections from control mice (**a,c**) and βV59M mice after 2 weeks diabetes (**b,d**). Above, immunofluoresence for insulin (green), glucagon (pink) and cleaved-caspase-3 (white). Below, PAS staining for glycogen (pink) and haematoxylin staining for nuclei (blue). Scale bars, 200 µm. (**b,d**) Insets top left show an 'empty' β-cell staining for glycogen, insulin and cleaved-caspase-3. Insets bottom right show insulin⁺-positive β-cells containing (arrowed) or lacking (arrowhead) both glycogen and cleaved-caspase-3. (**e,f**) Electron micrographs from 4-week diabetic βV59M mice using conventional fixation, which leads to loss of glycogen and electrolucent cytoplasm (**e**) or lead-citrate fixation, which preserves glycogen (**f**). Apoptotic bodies (Ap) filled with glycogen are evident in E (as electrolucent cytoplasm) and F (as glycogen granules, G). N, nucleus, *apoptotic nucleus. Scale bar 5 µm (**e**), 2 µm (**f**). (**g,h**) PAS-stained INS-1 cells following 48-h culture in 25 mM glucose without (**g**) or with 0.5 mM metformin (**h**). (**i**) Mean ± s.e.m. number of cleaved-caspase-3 positive INS-1 cells, expressed as a percentage of the total number of INS-1 cells, for cells cultured for 48-h at 25 mM glucose with (black bar), or without 0.5 mM metformin (white bar). $n = 25$ fields of view from three experiments. *$P < 0.05$ (t-test). (**j**) Quantitative PCR of Ppp1r3c mRNA from INS-1 cells cultured for 48-h in 5 mM glucose (black bar), and 25 mM glucose with (white bar) or without (checked bar) 0.5 mM metformin. Data are mean ± s.e.m. ($n = 3$). *$P < 0.05$ (t-test). (**k,l**) UGP2 mRNA (**k**) and protein (**l**) levels in INS-1 cells 72 h after treatment with control (siCONT) or UGP2 (siUGP2) siRNA and culture for 48-h at 25 mM glucose. Data are mean ± s.e.m. ($n = 3$). *$P < 0.05$. (**m,n**) Representative images of PAS-stained INS-1 cells 72 h after treatment with control (**m**) or UGP2 (**n**) siRNA and culture for 48-h at 25 mM glucose. Data are representative of three experiments. (**o**) Quantitative PCR of Ppp1r3c mRNA in INS-1 cells 72 h after treatment with control (black bars) or UGP2 (white bars) siRNA and culture for 48-h at 25 mM glucose. Data are mean ± s.e.m. ($n = 3$). *$P < 0.05$ (t-test). (**p**) Western blots of cleaved caspase-3 and vinculin (loading control) from INS-1 cells treated with control (siCONT) or UGP2 (siUGP2) siRNA 72 hrs previously and then cultured for 48-h at 25 mM glucose ($n = 3$).

observed following 72 h of knockdown. We had expected this to lower glycogen levels, because UGP2 catalyses the addition of UDP to glucose-1-phosphate to promote glycogen formation. Unexpectedly, however, it led to a paradoxical rise in glycogen at high glucose (Fig. 7m,n; Supplementary Fig. 8). We attribute this to the fact that *Ppp1r3c* expression was enhanced (Fig. 7o). Importantly, the elevated glycogen was accompanied by increased caspase-3 activation (Fig. 7p; Supplementary Fig. 7B).

Taken together, these studies demonstrate that caspase activation correlates with glycogen content, suggesting that excessive

glycogen accumulation may mediate, at least in part, enhanced β-cell death in response to chronic hyperglycaemia.

## Discussion

Our data show that hyperglycaemia results in a progressive impairment of β-cell carbohydrate metabolism and metabolic gene expression that begins within 24 h of exposure to high blood glucose (Fig. 8). As diabetes duration increases, insulin granule density decreases and glycogen accumulates to such an extent that

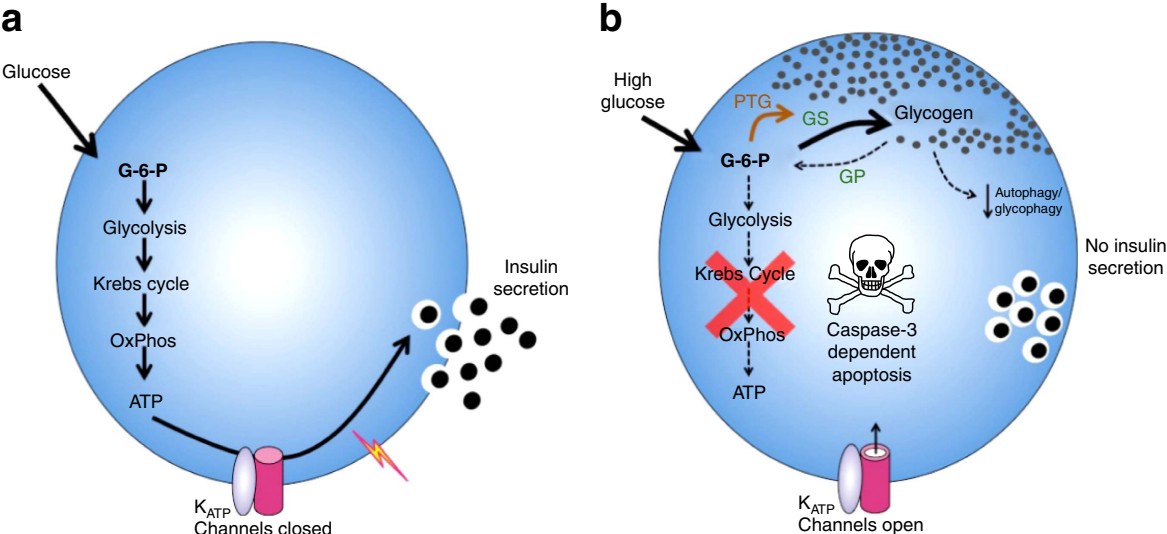

**Figure 8 | Suggested mechanism for β-cell dysfunction and apoptosis. (a)** In response to a rise in blood glucose, glucose enters the β-cell and is phosphorylated to glucose-6-phosphate (G6P). It is then metabolized via glycolysis and oxidative phosphorylation to produce ATP. ATP inhibits the ATP-sensitive $K^+$ channel ($K_{ATP}$) and thereby promotes membrane depolarization and insulin secretion. **(b)** Chronic hyperglycaemia leads to impaired oxidative metabolism and reduced ATP generation in response to glucose, thereby preventing $K_{ATP}$ channel closure, electrical activity and insulin secretion. Hyperglycaemia leads to elevation of G6P, which stimulates glycogen synthase and also leads to elevation of Ppp1r3c (PTG), both of which promote glycogen accumulation. Hyperglycaemia may also impair autophagy, further increasing glycogen storage and enhancing β-cell death. $K_{ATP}$, ATP-sensitive $K^+$ channel. G6P, glucose-6-phosphate. PTG, protein targeting to glycogen (Ppp1r3c). GS, glycogen synthase. GP, glycogen phosphorylase.

it causes gross distortion of β-cell ultrastructure. Changes in gene expression, metabolic pathways and impaired autophagy may contribute to glycogen accumulation. Studies of mutGCK mice further indicate that increased glucose metabolism, rather than glucose *per se*, drives glycogen synthesis and many of the changes in metabolic gene expression. Impaired autophagy and increased cleaved caspase-3 suggest apoptotic pathways are also stimulated. Similar changes to those seen in islets of βV59M mice *in vivo* were also observed in control mouse islets, in islets from non-diabetic human donors and in INS-1 cells exposed to hyperglycaemia *in vitro*.

Restoration of euglycaemia by glibenclamide rapidly restored normal glucose metabolism, dissipated glycogen stores, stimulated autophagy and inhibited apoptosis. This was more easily achieved after short-term hyperglycaemia. The progressive decline in β-cell structure, function and insulin content caused by chronic hyperglycaemia likely account for the reduced ability of βV59M mice with a longer duration of diabetes, and of older patients with neonatal diabetes[10], to respond to sulphonylurea therapy.

*In vivo*, hyperglycaemia is linked with hypoinsulinaemia, making it difficult to distinguish which is causative with certainty. However, because the insulin concentration of the culture medium was constant in our *in vitro* studies of islets and INS-1 cells, the changes we observed must be a function of hyperglycaemia. This is supported by experiments using partially pancreatectomized rats where expression of many genes important for glucose-stimulated insulin release, and a range of β-cell specific transcription factors, was altered by hyperglycaemia and restored by correction of blood glucose alone with phloridzin[2].

Analysis of a mouse model with an activating glucokinase mutation targeted to the β-cell[30] revealed that enhanced glucose metabolism, rather than glucose *per se*, drives glycogen formation. This is evident because mutGCK animals were normoglycaemic at the age at which we studied them. In β-cells exposed to high glucose[27], glucose-6-phosphate (G6P) increases rapidly because intracellular glucose tracks the extracellular concentration and glucokinase is a high capacity, high $K_m$ enzyme. Despite their

normal blood glucose levels, G6P is also expected to be elevated in mutGCK mouse β-cells, due to the enhanced glucokinase activity. Glucose-6-phosphate is a potent allosteric activator of glycogen synthase[17] and both glucose and G6P inhibit glycogen phosphorylase[35]. These allosteric effects on enzyme activity potentiate glycogen accumulation, ensuring that it is rapidly initiated when glucose increases and quickly reversed when glucose levels fall.

Glycogen stores are only evident when measures are taken to preserve glycogen during fixation and processing; if this is not done, glycogen loss leads to areas of unstructured cytoplasm that have been variously described as 'clear', 'empty' or 'vacuolated' at the light microscope level. Such 'empty β-cells' have been reported previously in patients with severe uncompensated diabetes[36] and in numerous more recent studies of pancreatic islets from diabetic mouse models[16,21,22,24,25,37]. These studies used fixation methods that were not designed to preserve glycogen, suggesting that the 'empty' cells observed may not, in fact, have been empty but contained glycogen that was lost during processing. This would also argue that altered metabolism and glycogen accumulation is a feature of all diabetic β-cells.

Small amounts of glycogen can be beneficial to a cell, by providing an energy store at times when the substrate supply falls. However, excessive glycogen may be deleterious: in other cell types it is associated with cell death. For example, glycogen accumulation in human diabetic nephropathy[38,39] is accompanied by enhanced apoptosis[40], and in neurones, glycogen accumulation triggers a decrease in autophagy that leads to cell death[41–43]. The marked increase in cleaved caspase-3 in glycogen-positive β-cells, and the gross distortion of β-cell ultrastructure we observed suggests excessive glycogen deposition may also be deleterious in β-cells.

In support of this idea, we found a strong positive correlation between the extent of cleaved caspase-3, the density of β-cell glycogen granules and the level of PTG (*Ppp1r3c*) expression that was independent of extracellular glucose. When the glucose concentration was held constant at 25 mM, metformin reduced

both PTG levels and glycogen accumulation, and led to a decrease in cleaved caspase-3. Conversely knockdown of UGP2 increased PTG and glycogen at 25 mM glucose further, while simultaneously elevating cleaved-caspase-3. This suggests that increased β-cell glycogen may lead to caspase-3 activation and subsequent β-cell apoptosis. Indeed, we observed a striking accumulation of glycogen in apoptotic bodies in βV59M islets. It is likely that the deleterious effect of glycogen on the β-cell varies with the extent of its accumulation, and thus with the duration and degree of diabetes; and that there is a threshold below which glycogen has no adverse action.

Currently it is widely assumed that β-cells do not store or metabolise glycogen. Indeed, a role for glycogen in normal β-cell function has recently been excluded by examining the effects of overexpression of PTG and knockdown of glycogen synthase I in mice[44]. However, this does not exclude a role for glycogen accumulation in diabetes[45]. Indeed, our data provide evidence that in combination with metabolic dysfunction and changes in autophagy, glycogen accumulation in severe uncompensated diabetes may promote β-cell loss via increased caspase-dependent apoptosis, and thereby enhance the hyperglycaemia. The presence of glycogen in islets from type-2 diabetic patients, further suggests that glycogen accumulation may contribute to β-cell failure in diabetes.

Glycogen accumulation implies a change in glucose handling by the β-cell. Our data demonstrate that chronic hyperglycaemia produces a dramatic change in β-cell metabolism, such that after 4 weeks of diabetes glucose fails to elevate NAD(P)H or ATP in βV59M islets. Failure of glucose to elevate NAD(P)H in diabetic mice carrying an activating Kir6.2 mutation has also been reported previously[46]. As NAD(P)H signals are principally mitochondrial in origin[47], our data indicate that hyperglycaemia impairs oxidative metabolism, which accounts for the reduction in glucose-stimulated ATP production and the diversion of glucose into glycogen. At least some of the marked changes we observed in the expression of key metabolic genes may underlie the altered ATP and NAD(P)H responses to glucose.

It is remarkable that, in vivo, all glycogen disappears from the β-cell within 24 h of normoglycaemia produced by sulphonylurea therapy, indicating glycogen breakdown is enhanced and/or glycogen synthesis is reduced. Both G6pc2 and Ppp1r3c are among the genes whose expression levels are rapidly restored by normoglycaemia. In combination with the reduced substrate concentration and restoration of normal metabolism (see below), these changes in gene expression are expected to lead to reduced glycogen synthesis when euglycaemia is restored. Our data also provide evidence that glycogen is broken down via autophagy when blood glucose is normalized. This has been demonstrated previously in other tissues[19], but not in β-cells.

Insulin was also able to reverse the diabetes-induced changes in islet insulin content and glycogen, but took longer to do so. This likely reflects the ability of sulphonylureas to promote membrane depolarization and calcium entry, which in turn facilitates activation of $Ca^{2+}$-dependent metabolic enzymes such as mitochondrial dehydrogenases and glycogen phosphorylase[48,49].

Importantly, the glucose-stimulated increases in NAD(P)H and ATP in βV59M islets, like the glycogen accumulation, were restored within 48 h of normoglycaemia. Yet despite the rapid restoration of oxidative metabolism, the expression levels of some metabolic genes were not normalized, even after 4 weeks of euglycaemia. Changes in expression of these specific genes therefore may not be the only factor regulating glucose-stimulated ATP production. The data also raise the possibility that hyperglycaemia induces a form of metabolic reprogramming, in which some gene expression changes are either not reversed, or require lower glucose concentrations or longer time periods, to do so.

It is noteworthy that all the hyperglycaemia-induced changes in gene expression in INS-1 cells were fully reversed on return to low glucose, whereas this was not the case for those caused by 4-weeks of diabetes in vivo. This may reflect the longer duration of diabetes in vivo, or the fact that the return glucose concentration was lower and more stable in the INS-1 cell experiments. The differences observed in cultured cells emphasize the importance of in vivo experiments. Chronic hyperglycaemia cannot be mimicked in vitro as INS-1 cells die when exposed to > 25 mM glucose for more than a few days. Similarly, cultured islets may become ischaemic in the islet core, due to the lack of vascular perfusion. Thus in both cases long-term changes in β-cell structure and function due to diabetes cannot be faithfully recapitulated in vitro.

Many of the changes seen in βV59M mice are also found in islets isolated from patients with type-2 diabetes (T2DM). Similar alterations in insulin content/staining[50–52] and expression of key metabolic genes (for example, fructose bisphosphatase[53]) have been reported. The marked decrease in glucose-stimulated ATP production we observe in βV59M islets is also seen in T2DM islets[54], and would be expected to impair insulin secretion. Although this cannot be demonstrated in βV59M islets (because $K_{ATP}$ channel activation prevents ATP-induced membrane depolarization) it likely contributes to impaired glucose-stimulated insulin release in T2DM. We also found significant glycogen stores in patients with T2DM who had documented poor glycaemic control. Early studies of islets isolated from patients with uncompensated diabetes (who died of diabetic coma) support this finding[36]. These data provide evidence that type 2 diabetes is associated with impaired β-cell metabolism that, at least in part, derives from the elevated blood glucose. Our results also reveal that, in mice, metformin has a protective effect on both glycogen accumulation and β-cell death induced by high glucose. It is possible that this may contribute to the efficacy of this drug in treating type-2 diabetes.

In conclusion, hyperglycaemia in βV59M mice induces profound metabolic dysfunction in β-cells. This is evidenced by progressive changes in metabolic gene expression profiles, decreased oxidative metabolism, impaired glucose-stimulated ATP production and glycogen accumulation. Prolonged exposure to hyperglycaemia also leads to loss of β-cells via caspase-dependent apoptosis. Studies on mice with an activating GCK mutation suggest that increased glucose metabolism drives most of these changes. Glycogen deposits were also found in the β-cells of patients with poorly controlled type-2 diabetes, and may contribute to impaired β-cell function in type-2 diabetes. Provided diabetes duration was short, restoration of euglycaemia dissipated glycogen stores and restored β-cell metabolism. However, many of the diabetes-induced changes in gene expression were not reversed. It is possible this will make the β-cell more susceptible to a subsequent metabolic insult. It was also more difficult to reverse diabetes of longer duration. Thus our data highlight the importance of good glycaemic control not only in patients with neonatal diabetes, but also those with type-2 diabetes.

## Methods

**Animal care.** All experiments were conducted in accordance with the UK Animals Scientific Procedures Act (1986) and University of Oxford ethical guidelines. Mice were housed in same-sex littermate groups of 2–8 animals, in a temperature and humidity controlled room on a 12-h light-dark cycle (lights on at 07:00). Regular chow food (63% carbohydrate, 23% protein, 4% fat; Special Diet Services, RM3) was freely available. Water was available at all times. Because untreated diabetic mice are polyuric, they were maintained on high absorbency bedding.

**Generation of genetically modified mice.** Mice expressing the Kir6.2-V59M transgene in insulin-secreting cells (βV59M mice) were generated using a Cre-lox approach, as previously described[3]. Expression was induced in 12–14 week-old

male and female mice with a mixed (C3H, C57BL/6, 129/sv) genetic background by a single subcutaneous injection of 0.4 ml of 20 mg ml$^{-1}$ tamoxifen in corn oil (Sigma). Hyperglycaemia was maintained for 2 or 4 weeks. Some animals were then implanted subcutaneously with either a slow-release glibenclamide pellet (Innovative Research of America) or a placebo pellet containing vehicle but no drug, under general anaesthesia (2% isoflurane, Animal Care Ltd.). Only mice in which normoglycaemia was restored within 48 h were used in the reversal studies, unless otherwise stated. Littermates (wild-type mice, RIPII-Cre-ER mice and mice expressing only the floxed Kir6.2-V59M gene[4]) were used as controls. Genotypes were identified by PCR using genomic DNA isolated from ear biopsies (DNeasy Blood and Tissue kit; QIAGEN). The presence of the Kir6.2-V59M mutant gene was confirmed using the following set of primers: P1, 5′-AAAGTCGCTCTGA GTTGTTATC-3′; P2, 5′-GATATGAAGTACTGGGCTCTT-3′; and P3, 5′-GCA TCGCCTTCTATCGCCT-3′. These amplify a 590-bp amplicon from the WT ROSA26 allele but a 460-bp product from the targeted allele. The presence of the RIP-Cre gene was confirmed by the amplification of an approximately 230-bp product, using forward ACGAGTGATGAGGTTCGCA and reverse ATGTTTAG CTGGCCCAAATGT primers. For both genes, PCR conditions were 94 °C for 3 min, followed by 45 cycles of 94 °C for 30 s, 57 °C for 45 s, 72 °C for 1 min 30 s. This was followed by a final extension at 72 °C for 10 min.

Mice expressing an activating mutation in the glucokinase gene (mutGCK mice) were generated as described previously[30].

**Microarray studies.** Mice were killed by cervical dislocation, the pancreas removed, and islets isolated by liberase digestion and handpicking[3]. Total RNA was extracted immediately using QIAshredders followed by the RNeasy Mini Kit (Qiagen), including an on-column DNase digestion step to remove traces of genomic DNA. RNA concentration was determined using a NanoDrop neonatal diabetes (ND)-1000 spectrophotometer (Thermo Scientific), and its integrity >6.0 verified with an Agilent 2100 BioAnalyzer.

Labelled sense ssDNA for hybridization was generated from 200 ng starting RNA with the Ambion WT Expression Kit, the Affymetrix WT Terminal Labelling and Controls Kit and the Affymetrix Hybridization, Wash, and Stain Kit, according to the manufacturer's instructions. Sense ssDNA was fragmented and the distribution of fragment lengths measured on a BioAnalyzer. The fragmented ssDNA was labelled and hybridized to the Affymetrix GeneChip Mouse Gene 1.0 ST Array (Affymetrix). Chips were processed on an Affymetrix GeneChip Fluidics Station 450 and Scanner 3000.

Microarray data were normalized using GeneSpring GX (Agilent). Differentially expressed genes were identified using Student's $t$-test with a $P$ value cut off of ≤0.05 and a fold change difference of ≥1.5 within GeneSpring. GO-Elite was employed to assess significantly regulated gene ontology (GO) terms in the data sets described above. At least 3 genes and ≥10% genes in a GO term needed to be changing with a permuted $P$ value of ≤0.05 to be included. Microarray data are available in the ArrayExpress database (www.ebi.ac.uk/arrayexpress under experiment E-MTAB-5086).

**RNAseq studies.** MutGCK mice were sacrificed 4 days after activation of the transgene, the pancreas removed and islets isolated by collagenase digestion followed by hand picking. RNA sequencing libraries were constructed from 200 ng of total RNA isolated using the TruSeq RNA sample prep kit (Illumina). Single read sequencing was performed on Illumina hiSeq2000 to 100 bp. Reads were aligned to the mouse genome mm9 using RUM[55]. Read counts for RefSeq transcripts were processed with EdgeR[56] to generate fold changes and $P$ values. $P$ values were converted to false discovery rates using the Benjamini & Hochberg mode of the R function p.adjust. Differentially expressed genes were identified using an FDR threshold of 10%. mRNA levels were expressed in reads per kilobase of transcript per million mapped reads (RPKM). RNAseq data are available in the GEO database (http://www.ncbi.nlm.nih.gov/geo accession number GSE86949).

**qPCR.** Total RNA was prepared from isolated islets using the RNeasy Lipid Tissue Mini Kit (Qiagen) with an on-column DNase digestion step to remove genomic DNA. RNA concentration was determined in triplicate using a NanoDrop ND-1000 spectrophotometer (Thermo Scientific) and quality checked with an Agilent Bioanalyzer. RNA was amplified using the Ovation Pico WTA System V2 (NuGen) kit and reversed transcribed. qPCR was performed using an Applied Biosystems StepOne Plus Real-Time PCR system. Each reaction consisted of 2 ng amplified cDNA, 0.5 μl Taqman probe, 10 μl Taqman Fast Advanced Master Mix, and nuclease-free water (final volume of 15 μl). Transcript levels of metabolic genes and the reference gene *Actb* quantified using Taqman probes (for probes see Supplementary Table 2). The reaction cycle comprised an initial denaturation for 10 min at 95 °C, followed by 40 cycles of 95 °C/15 s and 60 °C/60 s. All reactions were performed in triplicate. The StepOne Plus Real-Time PCR software (Applied Biosystems) was used to measure threshold cycle (Ct) values. Gene expression was determined using the Pfaffl method.

**Western blotting.** Proteins were extracted from INS-1 cells and islets isolated from control or diabetic βV59M mice in RIPA lysis buffer (ThermoScientific) containing protease inhibitor for 30 min on ice. Samples were then centrifuged for

15 min at 13,000 r.p.m. and 4 °C, and supernatants used for blotting. Extracts were separated on 4–12% SDS–polyacrylamide gels, blotted onto nitrocellulose membranes, and assayed using specific antibodies (listed in Supplementary Table 2). INS-1 cells were originally developed by Claes Wollheim (Geneva) and kindly supplied by Jochen Lang (Bordeaux).

**Serum lipid and glucose measurements.** Whole blood was extracted, left to stand at room temperature for 30 min, and serum was then collected following centrifugation at 5,000 $g$ for 30 min at 4 °C. Commercially available kits were used to measure HDL and LDL/VLDL cholesterol (Abcam; ab65390), triglycerides (Abcam; ab65336), free fatty acids (ab65341) and alanine aminotransferase activity (Sigma; MAK052). Serum glucose was measured using the Glucose (HK) Assay kit (Sigma-Aldrich) as per the manufacturer's instructions.

**Electron microscopy.** For routine structural analyses, pancreatic tissue or isolated islets were fixed in 2.5% glutaraldehyde in 0.1 M phosphate buffer for at least 1 h, post-fixed in 1% osmium tetroxide, dehydrated in graded ethanol, and embedded in Spurr's resin. Ultrathin sections (70 nm) were cut onto Ni$^{2+}$ grids, and, for standard morphology, contrasted with 2% uranyl acetate and lead citrate. They were examined in a Jeol 1010 microscope (Welwyn Garden City, UK) with an accelerating voltage of 80 kV and a digital camera (Gatan, Ca, USA).

Specimens for immunolabelling were not osmicated, but dehydrated in graded methanol and embedded in London Resin Gold resin (LRG) (Agar Scientific, Stansted, UK) which was cured in UV light at −20 °C overnight. Ultrathin sections cut onto nickel grids were washed in 0.1 M PBS containing egg albumin (Sigma, UK) and incubated with primary antibodies at effective concentrations (Supplementary Table 2). Antibody binding sites were identified using either protein A gold (10 or 15 nm) (Biocell, Cardiff, UK) or biotin conjugated antibodies (Dako) and streptavidin gold (10 or 15 nm) (Biocell). Sections were contrasted with 2% uranyl acetate and lead citrate and were examined in a Joel 1010 electron microscope fitted with a digital camera (Gatan, Ca, USA).

Intracellular glycogen particles are not osmiophilic, are soluble in uranyl acetate, and are not preserved by standard techniques[57,58]. To visualize glycogen, therefore, islet cells were post-fixed in a mixture of 1% osmium and a saturated solution of lead citrate (addition of 12.5% Reynolds lead citrate solution) in 0.1 M phosphate buffer for 1 h and processed as above into Spurr's resin. Glycogen was visualized using the periodic acid-thiosemicarbazide silver proteinate method (Vye MV and Fischman DA 1971); 1% thiosemicarbazide (dissolved in 25% acetic acid) and 1% silver proteinate (dissolved in water) (Elektron Technologies, Stanstead UK) were prepared freshly for each experiment. Grids were not treated with uranyl acetate or lead citrate for glycogen visualization.

Autophagosomes were defined as double-membrane vesicles containing electron-dense membranes and other cytoplasmic material (Fig. 5h).

**Immunostaining and morphometric analysis.** Immunostaining was performed on wax-embedded pancreatic sections fixed in 10% neutral-buffered formalin overnight at 4 °C. Dewaxed and rehydrated sections (5-μm thick) were blocked in 3% swine serum for 30 min at room temperature then incubated with primary antibodies overnight at 4 °C. Secondary fluorescent antibodies were incubated at room temperature for 1 h and counterstained with DAPI for nuclear visualisation. For antibodies, see Supplementary Table 3. PAS staining for glycogen was performed using the Abcam kit (ab150680). Human surgical and post-mortem pancreatic specimens were obtained with appropriate consent and with relevant UK local and national ethics permissions (1986) and have been approved by the University of Oxford.

**ATP measurements.** ATP levels were measured using the Roche Bioluminescence assay kit HS II. Isolated islets were washed briefly in glucose-free RPMI media plus 10% FCS and 1% penicillin/streptomycin. Batches of 10 islets were incubated in test solution for 2 h at 37 °C and then lysed using the kit buffer at 25 °C for 5 min with gentle shaking. ATP was measured according to the manufacturer's instructions.

**NAD(P)H measurements.** NAD(P)H autofluorescence from isolated islets was measured at 37 °C either immediately following isolation (for 4-week diabetic βV59M islets) or after 72h culture at different glucose concentrations (control islets). Islets were cultured in RPMI media plus 10% FCS and 1% penicillin/streptomycin at various glucose concentrations at 37 °C in a humidified atmosphere of 5% $CO_2$/95% air. Autofluorescence was excited at 350 nm and emission measured between 400 and 500 nm. During recording, islets were continuously superfused with a solution containing (mM); 132 NaCl, 5.6 KCl, 2.6 $CaCl_2$, 1.1 $MgCl_2$, 1.2 $NaH_2PO_4$, 4.2 $NaHCO_3$, 10 HEPES (pH 7.4) and glucose as indicated. In some cases, experiments were terminated by application of FCCP (4 μM), which decreased NAD(P)H fluorescence in both control and mutant islets (not shown). Traces were normalized to the signal obtained at the start of the recording in 1 mM glucose, to account for differences in islet size.

**INS-1 cell culture.** INS-1 832-13 cells were cultured at 37°C in a humidified atmosphere of 5% $CO_2$/95% air in RPMI (Gibco ref: 31870–025) supplemented with: 10% FBS (Ref-10270–106; Lot-41F6 245C), 50 µM β-mercaptoethanol, 1 mM sodium pyruvate, 10 mM HEPES, 1% pen/strep, 1% glutamax. The standard glucose concentration was 11 mM. For experiments, cells were seeded in 11 mM glucose for 24 h, and the media was then changed to 2 mM or 25 mM glucose for 24 h before harvesting. Knockdown of UGP2 was achieved using Lipofectamine RNAiMAX (Invitrogen, CA, USA) and 20 pmoles siRNA directed against rat UGP-2 (Invitrogen Silencer select RNAi). Stealth RNA interference siRNA Negative Control Medium GC (Invitrogen) was used as a control. Following knockdown for 72 h, cells were challenged for 48 h with 5 mM or 25 mM glucose and mRNA and protein extracted.

**Statistics.** Data are mean values ± s.e.m. of the indicated number of experiments. Significance was tested using one-way ANOVA, Mann–Whitney U Test, Student's *t*-test, and a Bonferroni *post hoc* test applied, as indicated.

**Data availability.** Microarray data are available in the ArrayExpress database (www.ebi.ac.uk/arrayexpress under experiment E-MTAB-5086). RNAseq data can be found in the GEO database with accession number GSE86949 (http://www.ncbi.nlm.nih.gov/geo). The authors declare that all data supporting the findings of this study are available within the paper (and its supplementary information files) or can be obtained from the authors upon reasonable request.

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

## Acknowledgements

We thank Art Hand and Mark Terasaki for pointing out the presence of glycogen in our diabetic islets, Katia Mattis and Raul Terron-Exposito for genotyping, the animal house staff for taking care of the mice, and Wellcome Trust Integrative Physiology Initiative in Ion Channels and Diseases of Electrically Excitable Cells (OXION) for use of its microarray facility. We thank the Wellcome Trust (grant nos 884655, 089795) and the European Union (ERC Advanced grant 322620) for support. The mutGCK work was supported by the ISF-JDRF Joint Program in Type 1 Diabetes Research (grant no. 1506/12). FMA holds an ERC Advanced Investigators award and a Royal Society Research Wolfson Merit Award. MFB holds a Wellcome Trust OXION Training Fellowship. MR is supported by a Novo Nordisk postdoctoral fellowship run in partnership with the University of Oxford.

## Author contributions

M.F.B. designed the study, conducted experiments and data analysis, and contributed to the writing of the manuscript. M.R. carried out experiments on INS-1 cells, western blotting and blood biochemistry, and contributed to the writing of the manuscript. M.I. quantified insulin staining and C.H. the area of unstructured cytoplasm. K.S. and S.L. performed the microarray and M.V.C. qPCR experiments. S.T.-B., D.D., B.G. and Y.D. performed the RNAseq of mutGCK mice and provided mutGCK mouse tissue sections. P.R. contributed to the NADH imaging, qPCR experiments and writing of the paper. A.C. supervised and contributed to the EM experiments. F.M.A. designed and supervised the project, and co-wrote the manuscript. All authors revised and approved the manuscript.

## Additional information

**Competing financial interests**: The authors declare no competing financial interests.

**How to cite this article**: Brereton, M. F. *et al.* Hyperglycaemia induces metabolic dysfunction and glycogen accumulation in pancreatic β-cells. *Nat. Commun.* **7,** 13496 doi: 10.1038/ncomms13496 (2016).

**Publisher's note**: 

