## [Peer Review File · Nature Communications]

Reviewers' comments:

Reviewer #1 (Remarks to the Author):

A. Summary: this manuscript entitled "Hyperglycemia induces metabolic dysfunction and glycogen accumulation in pancreatic b-cells" by Brereton et al. shows that prolonged hyperglycemia by activating Katp channel mutation or by human diabetes leads to accumulation of glycogen in pancreatic islets and b-cell dysfunction such as decreased ATP generation. Glycogen accumulation was claimed to cause b-cell dysfunction, which was again claimed to be due to defective autophagy.

B. Originality and interest: this paper seems to be novel in that glycogen accumulation in islet cells was studied. In vitro and in vivo models are also unique. This paper could be an attractive one to readers in the field of diabetes and glucotoxicity.

C. Data & methodology: some data are not so convincing. 1) To clearly demonstrate the role of autophagy in glycogen turnover, autophagic activity rather than autophagy level needs to be determined. 2) Also both the number of autophagosome and that of autophagolysosome need to be determined in Fig. 5E. 3) The mechanism of defective autophagy in 4-week-diabetic islets (despite accumulated glycogen) needs to be addressed in more detail besides beclin-1 expression. 4) The method detecting caspase-3 in white color needs to be explained in more detail. 5) In Fig. 6B, many islet cells show strong staining of active caspase-3 staining (white color). However, the authors state that there was no obvious increase of apoptotic islet cells in their previous studies by other studies. It is hard to understand why staining for active caspase-3 is so strong. 6) In Fig. 7J, a more direct evidence showing colocalization of glycogen with lysosome needs to be presented such as colocalization with lysosomal markers, etc. 7) The authors suggest that defective islet cell function (e.g. defective ATP generation) is caused by the accumulation of glycogen accumulation. Some more cellular or biochemical mechanism needs to be addressed in addition to the changes of ultrastructure and decreased insulin content. 8) Accumulation of glycogen is critically depends on insulin. Insulin level and its impact need to be determined (e.g. in the case of INS-1 cells that produces insulin). Was insulin release from bV69M islet cells determined in this study, while the authors state that insulin concentration of the culture medium was constant in in vitro experiment? 9) Some biochemical determination of intracellular glycogen level will be helpful to convince other investigators.

D. Appropriate use of statistics and treatment of uncertainties: no specific problem.

E. Conclusions: More data are necessary to support the authors' conclusion conclusively.

F. Suggested improvements: 1) Determination of autophagic flux in addition to autophagy level 2) Counting of autophagolysosomes 3) Use of lysosomal markers to study colocalization with lysosomes 4) More mechanistic studies regarding defective autophagy 5) More mechanistic studies regarding b-cell dysfunction due to glycogen accumulation 6) Measurement of ambient insulin level or insulin release and its impact on glycogen 7) Biochemical assay of glycogen accumulation

G. References: no specific problem

H. Clarity and context: no specific problems except some missing data to support the authors' logic

Reviewer #2 (Remarks to the Author):

Review Brereton et al

This both novel and important study explores a long unresolved issue in diabetes and pancreatic beta-cell research as to whether carbohydrates are stored as glycogen in the beta-cell or not.

The authors address this issue intelligently using an inducible and beta-cell specific mouse model for neonatal diabetes with an activating Katp channel mutation, which allows controlled onset of hyperglycemia and its reversal by pharmacological means, in combination with an array of mostly well-chosen methods.

The overall observation that carbohydrates are stored in the form of glycogen in beta-cells under diabetic conditions is strongly supported by the data and statistical analysis. As to how this accumulation is regulated and which the functional consequences are, there are a few issues that are worthy of deeper exploration.

Feedback

1. The faster reversal of glycogen accumulation by sulphonylureas as compared to insulin treatment suggests the involvement of Ca²⁺-initiated cellular signals in glycogen breakdown (as also supported by the data in Fig. 3C and pointed out by the authors in the discussion). However, the fact that INS-1 cells with intact Ca²⁺-signaling readily accumulate glycogen is in this respect somewhat puzzling. There are several potential explanations, but repeating the same experiment, but in the presence of either a Katp channel opener, a Ca²⁺ channel blocker or Ca²⁺-free conditions, would help to clarify the situation.

2. Measuring autophagy is not straightforward and all methods have limitations. The authors have chosen (Fig. 5) mostly observational imaging assays. The data in Fig. 3E would benefit from being supported by more quantitative methods such as immunoblotting of Lcb3, preferably in the presence or autophagic flux inhibitors or enhancers.

3. A difficult and perhaps even impossible experiment to do: it would be interesting to learn the effect of culturing human islets in high glucose on glycogen accumulation and its reversal by sulphonylurea treatment, as well as exploring the effects of sulphonylurea treatment in human islets from donors with T2D.

4. There is in the ms great emphasis on the connection between glycogen accumulation and apoptotic signals. For understanding the development of the human disease, in which the time course is expected to be much slower than in these in vitro studies, it would be very informative to learn if/how the functional properties of glycogen-laden INS-1 cells differ from their control counterparts, i.e. Katp channel regulation, Ca²⁺ I-Vs and insulin release (either hormone release or single-cell techniques).

5. Minor: Were the qPCR experiments performed using only one reference gene?

The manuscript is very well written and should be easy to understand also by non-specialists.

Reviewer #3 (Remarks to the Author):

In a previous publication, Brereton and colleagues have reported an interesting new diabetes mouse transgenic model in which the human mutated Kir6.2-V59M gene is overexpressed via tamoxifen injection specifically in beta-cells. The model permits to rapidly turn off insulin secretion following Katp activation and to restore it by sulphonylurea (SU). Thus, the model allows to induce hyperglycemia and to reverse it on demand. Using this model they previously investigated (Nature Commun 2014) the effects of chronic hyperglycemia (4 weeks only) on islet structure and function. In the current follow-up study, the same model was used to study the progressive damage (24h, 2 and 4 weeks) induced by hyperglycemia (glucotoxicity) on islets. The authors have shown that hyperglycemia rapidly (24h) alters glucose metabolism by reducing glucose stimulated ATP production. Prolonged hyperglycemia (2 & 4 weeks) decreases insulin content, and is associated with the accumulation of

glycogen and some apoptosis. Reversal of hyperglycemia by SU is associated with islet autophagy and the rapid disappearance of glycogen reserves. In an vitro study using INS1 cells and isolated islets, the authors showed similar effects of hyperglycemia on the expression of genes, metabolism and glycogen accumulation. Glycogen accumulation was also shown in T2D islets. The authors conclude that the duration of hyperglycemia influences how these metabolic and gene expression changes are reversed by SU. The longer the hyperglycemia, the fewer animals recover with SU. They noticed also that even if euglycemia is achieved several changes in gene expression were not reversed suggesting that this could make beta-cells more susceptible to subsequent insults.

The manuscript is well written, the analyses are appropriate and statistics are adequate and well used. The study is the logical continuation of their previous work published in the same journal. However, as it stands, the study is largely descriptive, in part incomplete, some controls are missing and several of the conclusions are not enough supported by the data. Also, reliable glycogen quantification has not been performed thus it is difficult to conclude about a role of glycogen accumulation following hyperglycemia. Finally, the work is not really novel. The title states "hyperglycemia induces metabolic dysfunction and glycogen accumulation in pancreatic β -cell" but these glucotoxicity effects are largely known since many years with multiple studies by many groups having documented this in normal and tumoral β -cell lines.

Additional major comments

Study is incomplete, some controls are missing and several of the conclusions are not enough supported by the data.

1-The proof that the reversibility depends on the hyperglycemic period, is not optimal (Fig 2). The authors convincingly demonstrate that the beta cell mass (%ins+ area) and islet density gradually decreased with the duration of hyperglycemia (Fig 1). They mention that 88% of the animals that were hyperglycemic for 2w were euglycemic after SU treatment vs 48% for animals that have been hyperglycemic for 4 weeks. Treatment with SU after 2w under hyperglycemia restored insulin content (Figure 2B), however, the authors did not show the effect of SU after 1 and 4 weeks under hyperglycemia. Moreover, they did not show the effects of SU on the beta-cell mass and islets density islets. Also, they did not show insulin content, beta-cell mass and islet density in the animals that remained hyperglycemic. Is the normalization of the insulin content only due to the restoration of euglycemia or due to the SU treatment per se or via an indirect effect of SU on other organs (Fig 2B)? This reasoning is supported by Figure 3C results showing that islets incubated at normal glucose concentration in the presence of SU have much increased glucose stimulated NADH levels compared to islets incubated at normal glucose without SU. Thus, normal glucose alone does not fully restore glucose stimulated NADH levels, suggesting that SU might do something independently of glucose levels. Authors should show the reversibility of insulin content of islets, beta-islet cell mass and density after 1, 2 and 4 weeks under hyperglycemia in animals that become euglycemic after SU treatment and in animal that do not.

2- Similarly, the authors have shown that 4w hyperglycemia altered glucose stimulated NADH and ATP levels (Fig 3) in isolated islets. They also showed the reversibility induced by SU on glucose stimulated ATP and NADH levels in islet from animals that regain normal glycemia. What happens in islets from animals that stayed hyperglycemic? This would be a control. And what would happen in islet from animals that were hyperglycemic for only 2w? Again the proof that the reversibility depends on the hyperglycemic period, is not optimal since here authors do not assess the 2w time point. The authors should show changes of glucose-stimulated ATP and NADH levels in animals under hyperglycemia for 24h, 2w and 4w and in animals that regain euglycemia after SU treatment (after 2 and 4w under hyperglycemia) and in animal that do not regain euglycemia. Without this, authors cannot conclude

that hyperglycemia alone induces metabolic dysfunction in beta-cells.

3- Interestingly, authors found that beta-cell glycogen levels increased along with hyperglycemia period, as assessed by histochemistry. However, they presented only few examples of islets with glycogen in them. No reliable quantification was performed. This is not quantitative (morphometric) assessment of the defect and glycogen should be directly measured. Also the presence of glycogen in islets of type 2 diabetic has been done by showing one islet from one donor (Fig 7a). Nothing can be concluded with n=1. Furthermore, it is not even sure that the presence of glycogen was in fact in beta-cells since authors did not confirm the identity of cells having glycogen. Authors should quantify islet glycogen content in isolated islets, show that contents are increasing with hyperglycemia period and that glycogen levels remain elevated in SU treated animal that stayed hyperglycemic and decrease in animals that regain euglycemia. Without this, authors cannot conclude that hyperglycemia induces glycogen accumulation in beta-cells.

Minor comments

4- Authors use the word diabetes to refer to hyperglycemia (examples line 5 of abstract; line 9 & 11, page 4). Diabetes represents a set of clinical signs and thus is more than just hyperglycemia. It is suggested that authors replace diabetes for hyperglycemia throughout the manuscript.

5- Title of the Y axis in Fig 2B should be "Insulin content..." not islet content.

Reply to Reviewer #1

A. Summary: this manuscript entitled "Hyperglycemia induces metabolic dysfunction and glycogen accumulation in pancreatic b-cells" by Brereton et al. shows that prolonged hyperglycemia by activating Katp channel mutation or by human diabetes leads to accumulation of glycogen in pancreatic islets and b-cell dysfunction such as decreased ATP generation. Glycogen accumulation was claimed to cause b-cell dysfunction, which was again claimed to be due to defective autophagy. Originality and interest: this paper seems to be novel in that glycogen accumulation in islet cells was studied. In vitro and in vivo models are also unique. This paper could be an attractive one to readers in the field of diabetes and glucotoxicity. Data & methodology: some data are not so convincing.

1) To clearly demonstrate the role of autophagy in glycogen turnover, autophagic activity rather than autophagy level needs to be determined.

We have now measured autophagic activity by LC3B Western blot (Fig. 5E). We show that LC3B-II protein is increased in 4-week diabetic β V59M mice. We also show an increase in p62 protein levels by Western blotting. This supports our previous observations of ubiquitin and p62 accumulation (using immunofluorescence and immunoelectron microscopy, respectively) in diabetic mice. The fact that after 4 weeks of diabetes we see accumulation of p62 and of LC3B-II, but no increase in the number of autophagosomes, suggests autophagy may be blocked or overwhelmed at some level downstream of LC3B activation.

2) Also both the number of autophagosome and that of autophagolysosome need to be determined in Fig. 5E.

We could clearly observe many of the double-membrane structures of autophagosomes when glycaemia was restored. However, we were unable to obtain specific immunogold labelling of lysosomes (using cathepsin B or LAMP2), and identifying autophagolysosomes by morphological parameters proved too subjective. We point out, however, that analysis of the regulation of glycogen levels by autophagy is beyond the scope of the current study, which focuses on the characterisation of β -cell glycogen in diabetes. The role of β -cell glycophagy will be the subject of a subsequent study.

3) The mechanism of defective autophagy in 4-week-diabetic islets (despite accumulated glycogen) needs to be addressed in more detail besides beclin-1 expression.

We now provide evidence that autophagy (as assessed by LC3B-II Western blot) is altered in diabetic β V59M islets. We also show increased p62 protein by Western blotting. This supports our previous data showing accumulation of ubiquitin and p62 in diabetic β V59M β -cells by electron microscopy.

4) The method detecting caspase-3 in white color needs to be explained in more detail.

This has now been clarified in the text.

5) In Fig. 6B, many islet cells show strong staining of active caspase-3 staining (white color). However, the authors state that there was no obvious increase of apoptotic islet cells in their previous studies by other studies. It is hard to understand why staining for active caspase-3 is so strong.

In our previous (2014) paper, we assessed apoptosis in the islets from diabetic β V59M mice by qualitatively assessing nuclear morphology from electron micrographs. These studies were inevitably limited to a small subset of β -cells (n=134) and thus we may have underestimated the extent of apoptosis, especially as nuclear chromatin condensation has a limited lifetime and is often missed. Furthermore, chromatin condensation is a late step in the apoptotic cascade and it is therefore possible that many more cells than we documented were undergoing earlier stages of the apoptotic process. This may explain why we observe significant cleaved caspase-3 staining, as this is an early marker of apoptosis.

6) In Fig. 7J, a more direct evidence showing colocalization of glycogen with lysosome needs to be presented such as colocalization with lysosomal markers, etc.

We used the periodic acid-schiff (PAS) method to detect glycogen. This fixation and staining procedure does not permit direct co-localisation of glycogen with antibody-based staining methods. However, as shown in Fig S5, INS-1 cells treated with a lysosomal inhibitor exhibit marked accumulation of LAMP2-positive lysosomes. These structures display a similar punctate pattern to glycogen staining. This would be consistent with the presence of glycogen in lysosomes.

7) The authors suggest that defective islet cell function (e.g. defective ATP generation) is caused by the accumulation of glycogen accumulation. Some more cellular or biochemical mechanism needs to be addressed in addition to the changes of ultrastructure and decreased insulin content.

This is not, in fact, what we state. We show that there is a correlation between glycogen accumulation and impaired metabolism. We do not imply that glycogen itself impairs metabolism – indeed, we consider this is unlikely. More plausible is that impaired oxidative metabolism leads to increased accumulation of G6P, which stimulates glycogen synthase. We apologise if our text was misleading and we have revised it to make it clearer.

8) Accumulation of glycogen is critically depends on insulin. Insulin level and its impact need to be determined (e.g. in the case of INS-1 cells that produces insulin). Was insulin release

from bV69M islet cells determined in this study, while the authors state that insulin concentration of the culture medium was constant in in vitro experiment?

In diabetic β V59M mice, plasma insulin levels are very low (often unmeasurable), (see Brereton et al 2014). Thus the glycogen accumulation that we observe is independent of insulin. There are two reasons for this.

First, G6P is a profound activator of glycogen synthase (Villar-Palasi & Guinovart, 1977; FASEBJ. 11, 544-558). In β -cells (unlike many other tissues) glucose uptake is independent of insulin, and thus elevation of plasma glucose leads directly to elevation of intracellular glucose and G6P, and thus to activation of glycogen synthase and glycogen accumulation. In islets, G6P can rise to very high levels (Matschinsky & Ellerman 1968, J Biol Chem 243, 2730-2736).

Secondly, we observed a marked increase in expression of Ppp1r3c in response to chronic hyperglycaemia. This enzyme (protein targeting to glycogen or PTG) plays a key role in glycogen synthesis, and its upregulation leads to glycogen storage in β -cells even at low glucose levels (Mir-Coll et al, 2016, Diabetologia 59, 1012-1020). In combination with high glucose (and high G6P), glycogen synthesis will be profoundly increased. PTG both enhances glycogen synthesis and suppresses glycogen breakdown (Greenberg et al, 2006; Mol Cell Biol 26, 334).

9) Some biochemical determination of intracellular glycogen level will be helpful to convince other investigators.

We were unable to develop a convincing and reproducible biochemical assay for glycogen content in islets or INS1 cells, despite extensive optimisation of several methodologies. All glycogen levels recorded directly mirrored the external glucose concentration in a time- and concentration-dependent manner. This suggests significant contamination from free glucose within the cell. To circumvent this problem, we counted individual glycogen granules in electron micrographs of INS1 cells, and the area of unstructured cytoplasm in β V59M islets (which indicates the presence of glycogen that has been washed out during fixation) (Fig.1E,6I). Electron microscopy studies also unequivocally confirm the presence of glycogen, rather than contaminating glycosylated proteins, as individual glycogen rosette granules can be identified.

D. Appropriate use of statistics and treatment of uncertainties: no specific problem.

E. Conclusions: More data are necessary to support the authors' conclusion conclusively.

F. Suggested improvements: (see below)

G. References: no specific problem

H. Clarity and context: no specific problems except some missing data to support the authors' logic

F. Suggested improvements: replies

1) Determination of autophagic flux in addition to autophagy level

2) Counting of autophagolysosomes

3) Use of lysosomal markers to study colocalization with lysosomes

4) More mechanistic studies regarding defective autophagy

- 5) *More mechanistic studies regarding b-cell dysfunction due to glycogen accumulation*
- 6) *Measurement of ambient insulin level or insulin release and its impact on glycogen*
- 7) *Biochemical assay of glycogen accumulation*

We have complied with the reviewer's suggestions, and our comments are given individually above.

Reply to Reviewer #2

This both novel and important study explores a long unresolved issue in diabetes and pancreatic beta-cell research as to whether carbohydrates are stored as glycogen in the beta-cell or not. The authors address this issue intelligently using an inducible and beta-cell specific mouse model for neonatal diabetes with an activating Katp channel mutation, which allows controlled onset of hyperglycemia and its reversal by pharmacological means, in combination with an array of mostly well-chosen methods. The overall observation that carbohydrates are stored in the form of glycogen in beta-cells under diabetic conditions is strongly supported by the data and statistical analysis. As to how this accumulation is regulated and which the functional consequences are, there are a few issues that are worthy of deeper exploration.

We thank the reviewer for their kind words.

1. The faster reversal of glycogen accumulation by sulphonylureas as compared to insulin treatment suggests the involvement of Ca²⁺-initiated cellular signals in glycogen breakdown (as also supported by the data in Fig. 3C and pointed out by the authors in the discussion). However, the fact that INS-1 cells with intact Ca²⁺-signaling readily accumulate glycogen is in this respect somewhat puzzling. There are several potential explanations, but repeating the same experiment, but in the presence of either a KATP channel opener, a Ca²⁺ channel blocker or Ca²⁺-free conditions, would help to clarify the situation.

We now show that human islets from non-diabetic patients, which presumably have intact calcium signalling, also accumulate glycogen when exposed to high glucose. We also now provide evidence that mice with an activating glucokinase mutation accumulate glycogen, even at normal glucose levels: these mice also have intact calcium signalling and in fact hypersecrete insulin. In our view, any effect of calcium on glycogen breakdown is far outweighed by the high levels of intracellular G6P, which both stimulates glycogen synthase and inhibits glycogen phosphorylase. When glucose (and/or G6P) is lowered, glycogen breakdown may vary according to whether calcium is present or not.

2. Measuring autophagy is not straightforward and all methods have limitations. The authors have chosen (Fig. 5) mostly observational imaging assays. The data in Fig. 3E would benefit from being supported by more quantitative methods such as immunoblotting of Lcb3, preferably in the presence or autophagic flux inhibitors or enhancers.

We have now performed immunoblotting for LC3B and p62 in β V59M islets (Fig. 5E), which supports the idea that autophagy is impaired in the diabetic state.

3. A difficult and perhaps even impossible experiment to do: it would be interesting to learn the effect of culturing human islets in high glucose on glycogen accumulation and its reversal by sulphonylurea treatment, as well as exploring the effects of sulphonylurea treatment in human islets from donors with T2D.

We now include data from human islets cultured at high and low glucose for 48 hours (Fig. 6C-E), which show marked glycogen accumulation at high glucose. We also show that these glycogen stores dissipate upon subsequent culture at low glucose. We have extended our studies on islets from human organ donors with type-2 diabetes to a total of 12 patients. Our cohort includes donors on sulphonylurea, insulin and metformin therapy, and varying degrees of glycaemia. However, given the small sample size, and varying levels of glycaemic control, we are unable to draw any conclusion as to how these therapies affect glycogen levels in humans.

4. There is in the ms great emphasis on the connection between glycogen accumulation and apoptotic signals. For understanding the development of the human disease, in which the time course is expected to be much slower than in these in vitro studies, it would be very informative to learn if/how the functional properties of glycogen-laden INS-1 cells differ from their control counterparts, i.e. Katp channel regulation, Ca²⁺ I-Vs and insulin release (either hormone release or single-cell techniques).

Apoptosis due to glycogen accumulation (in vivo) is a consequence of chronic (not acute) hyperglycaemia. We have extended our studies on the relationship between glycogen accumulation and apoptosis, and now show a strong correlation between glycogen levels and caspase 3 levels, that is independent of the ambient glucose level (Fig.7G-P).

There are many papers showing that culture of INS-1 cells at high glucose leads to impaired insulin secretion and altered Ca²⁺ homeostasis. For example, 48hr culture at 17mM glucose decreases secretion by ~30% (Gohring et al, 2013; J Biol Chem 289, 3786), and the L-type Ca²⁺ channel blocker nifedipine protects from high-glucose-induced endoplasmic reticulum stress and apoptosis (Wang et al, 2011; Int J Mol Sci 11, 7569). These culture conditions are similar to those that led to glycogen accumulation in our experiments. Our data suggests a link between glycogen levels and apoptosis. However, extensive additional studies are required to determine the precise mechanisms by which glycogen directly exerts its effects on beta-cell Ca²⁺ homeostasis and insulin secretion. This is beyond the scope of the current manuscript.

5. Minor: Were the qPCR experiments performed using only one reference gene?

We used only one reference gene (*Actb*), and its expression was not changed by any of the manipulations. This is now stated more clearly in the Methods. We chose not to use GADPDH as a reference gene, as it lies within the glycolytic pathway and our microarray studies suggested that its expression was affected by hyperglycaemia (although this was not significant).

The manuscript is very well written and should be easy to understand also by non-specialists.

Reply to Reviewer #3

In a previous publication, Brereton and colleagues have reported an interesting new diabetes mouse transgenic model in which the human mutated Kir6.2-V59M gene is overexpressed via tamoxifen injection specifically in beta-cells. The model permits to rapidly turn off insulin secretion following Katp activation and to restore it by sulphonylurea (SU). Thus, the model allows to induce hyperglycemia and to reverse it on demand. Using this model they previously investigated (Nature Commun 2014) the effects of chronic hyperglycemia (4 weeks only) on islet structure and function.

In the current follow-up study, the same model was used to study the progressive damage (24h, 2 and 4 weeks) induced by hyperglycemia (glucotoxicity) on islets. The authors have shown that hyperglycemia rapidly (24h) alters glucose metabolism by reducing glucose stimulated ATP production. Prolonged hyperglycemia (2 & 4 weeks) decreases insulin content, and is associated with the accumulation of glycogen and some apoptosis. Reversal of hyperglycemia by SU is associated with islet autophagy and the rapid disappearance of glycogen reserves. In an vitro study using INS1 cells and isolated islets, the authors showed similar effects of hyperglycemia on the expression of genes, metabolism and glycogen accumulation. Glycogen accumulation was also shown in T2D islets. The authors conclude that the duration of hyperglycemia influences how these metabolic and gene expression changes are reversed by SU. The longer the hyperglycemia, the fewer animals recover with SU. They noticed also that even if euglycemia is achieved several changes in gene expression were not reversed suggesting that this could make beta-cells more susceptible to subsequent insults.

The manuscript is well written, the analyses are appropriate and statistics are adequate and well used. The study is the logical continuation of their previous work published in the same journal. However, as it stands, the study is largely descriptive, in part incomplete, some controls are missing and several of the conclusions are not enough supported by the data. Also, reliable glycogen quantification has not been performed thus it is difficult to conclude about a role of glycogen accumulation following hyperglycemia. Finally, the work is not really novel. The title states "hyperglycemia induces metabolic dysfunction and glycogen accumulation in pancreatic β -cell" but these glucotoxicity effects are largely known since many years with multiple studies by many groups having documented this in normal and tumoral β -cell lines.

Additional major comments: Study is incomplete, some controls are missing and several of the conclusions are not enough supported by the data.

1-The proof that the reversibility depends on the hyperglycemic period, is not optimal (Fig 2). The authors convincingly demonstrate that the beta cell mass (%ins+ area) and islet density gradually decreased with the duration of hyperglycemia (Fig 1). They mention that 88% of the animals that were hyperglycemic for 2w were euglycemic after SU treatment vs 48% for animals that have been hyperglycemic for 4 weeks. Treatment with SU after 2w under hyperglycemia restored insulin content (Figure 2B), however, the authors did not show the effect of SU after 1 and 4 weeks under hyperglycemia. Moreover, they did not show the

effects of SU on the beta-cell mass and islets density islets. Also, they did not show insulin content, beta-cell mass and islet density in the animals that remained hyperglycemic.

Is the normalization of the insulin content only due to the restoration of euglycemia or due to the SU treatment per se or via an indirect effect of SU on other organs (Fig 2B)? This reasoning is supported by Figure 3C results showing that islets incubated at normal glucose concentration in the presence of SU have much increased glucose stimulated NADH levels compared to islets incubated at normal glucose without SU. Thus, normal glucose alone does not fully restore glucose stimulated NADH levels, suggesting that SU might do something independently of glucose levels.

Insulin therapy *in vivo* is able to both prevent and reverse the effects of hyperglycaemia. This was shown in our earlier study (Brereton 2014), where we found that insulin therapy was able to prevent the reduction in the percentage area of the islet staining for insulin as effectively as glibenclamide. Here, we show that insulin is able to restore β -cell granulation and glycogen accumulation (Fig.2). Thus euglycaemia (and restoration of insulin levels) is sufficient to reverse insulin content and β -cell ultrastructure. Although insulin was not as effective as glibenclamide, this might simply be a consequence of the fact that glucose homeostasis is not controlled as effectively by insulin (see Fig 1, Brereton et al 2014). By contrast, the blood glucose is very stable on sulphonylurea treatment.

We suggest that the greater NADH signal seen in islets subject to 4-weeks diabetes followed by incubation at low glucose and gliclazide, when compared with diabetic islets subsequently incubated at low glucose alone, is due to the membrane depolarization produced by gliclazide. This is expected to activate Ca^{2+} -dependent mitochondrial dehydrogenases and stimulate metabolism.

Authors should show the reversibility of insulin content of islets, beta-islet cell mass and density after 1, 2 and 4 weeks under hyperglycemia in animals that become euglycemic after SU treatment and in animal that do not.

Unfortunately we do not have tissues from these animals to perform the suggested analyses and it would take extensive and prolonged breeding to obtain sufficient numbers for rigorous studies. Furthermore, we do not think the data would add substantially to the manuscript. In addition, it has already been shown that animals with a (different) activating K_{ATP} channel mutation in which euglycaemia is only partially restored by insulin therapy also only show partial restoration of their islet insulin content, architecture and beta-cell mass (Wang et al, 2014; Cell Metabolism 19, 872).

2- Similarly, the authors have shown that 4w hyperglycemia altered glucose stimulated NADH and ATP levels (Fig 3) in isolated islets. They also showed the reversibility induced by SU on glucose stimulated ATP and NADH levels in islet from animals that regain normal glycemia. What happens in islets from animals that stayed hyperglycemic? This would be a control. And what would happen in islet from animals that were hyperglycemic for only 2w? Again the proof that the reversibility depends on the hyperglycemic period, is not optimal since here authors do not assess the 2w time point.

The authors should show changes of glucose-stimulated ATP and NADH levels in animals under hyperglycemia for 24h, 2w and 4w and in animals that regain euglycemia after SU

treatment (after 2 and 4w under hyperglycemia) and in animal that do not regain euglycemia. Without this, authors cannot conclude that hyperglycemia alone induces metabolic dysfunction in beta-cells.

Unfortunately it is not possible to perform all the experiments the reviewer requests due to the limited number of genetically modified animals it is possible to obtain, and the restrictions of our Home Office Licence, which limits the number of animals that can be used. Please see comment above.

Nevertheless, we have compared the effects of diabetes for 24h and 4wks. We show that significant changes in NADH and ATP begin to occur within just 24h of hyperglycaemia, both in vivo and in vitro. It is unlikely that intermediate durations of diabetes will produce novel changes that reach statistical significance, given the small changes in the NADH and ATP responses we observe for diabetes of 24h and 4wks duration. We also point out that the effects of hyperglycaemia on ATP and NADPH can be reversed following 4-wks of diabetes. Thus, it would be surprising if they were not also reversed after 24 hrs or 2 wks of diabetes.

3- Interestingly, authors found that beta-cell glycogen levels increased along with hyperglycemia period, as assessed by histochemistry. However, they presented only few examples of islets with glycogen in them. No reliable quantification was performed. This is not quantitative (morphometric) assessment of the defect and glycogen should be directly measured. Also the presence of glycogen in islets of type 2 diabetic has been done by showing one islet from one donor (Fig 7a). Nothing can be concluded with n=1. Furthermore, it is not even sure that the presence of glycogen was in fact in beta-cells since authors did not confirm the identity of cells having glycogen.

We have now performed quantification of glycogen in β V59M islets (Fig.1E) and INS1 cells (Fig.6H), as detailed above.

Our original manuscript included representative pictures from three type 2 diabetic donors, as was stated in the figure legend. We have now extended these studies to 9 patients and found glycogen in all of them (to varying extents). The identity of the human β -cells can be reliably determined from the fact that they express insulin, as shown by the presence of insulin granules in electron microscopy studies (Fig. 6B). Insulin granules are identifiable by the halo that surrounds the insulin (giving them a poached egg appearance).

Authors should quantify islet glycogen content in isolated islets, show that contents are increasing with hyperglycemia period and that glycogen levels remain elevated in SU treated animal that stayed hyperglycemic and decrease in animals that regain euglycemia. Without this, authors cannot conclude that hyperglycemia induces glycogen accumulation in beta-cells.

We have now quantified glycogen at the ultrastructural level and confirmed the relationship between circulating glucose levels and glycogen accumulation in β V59M islets (Figs. 1 and 6H). As stated above, we were unable to develop a convincing and reproducible biochemical assay for glycogen content measurements despite extensive optimisation of several methodologies. This was due to contamination with free glucose. However, we feel

that counting glycogen granules is a more accurate, albeit more laborious method of quantification as glycogen can be unequivocally identified.

Minor comments

4- Authors use the word diabetes to refer to hyperglycemia (examples line 5 of abstract; line 9 & 11, page 4). Diabetes represents a set of clinical signs and thus is more than just hyperglycemia. It is suggested that authors replace diabetes for hyperglycemia throughout

We respectfully disagree with this suggestion. While hyperglycaemia is the correct term when studying the effect of glucose on isolated cells and tissues, it is more appropriate to use diabetes when referring to the animal studies. Our mice have diabetes (high blood glucose, low insulin) that resembles that of patients with neonatal diabetes or type 1 diabetes.

5- Title of the Y axis in Fig 2B should be "Insulin content..." not islet content.

We thank the referee for pointing this out. It has been corrected.

REVIEWERS' COMMENTS:

Reviewer #1 (Remarks to the Author):

This revised manuscript entitled "Hyperglycemia induces metabolic dysfunction and glycogen accumulation in pancreatic b-cells" by Brereton et al. has been improved by the incorporation of reviewers' suggestions. I have now only a couple of minor inquiries.

1. In page 9, autophagic bodies were mentioned. There is no clear definition of autophagic bodies or mention about how to count autophagic bodies. Are 'autophagic bodies' same as autophagosomes?
2. In Fig. 7J, Pcc1r3c expression in the presence of normal glucose (rather than 25 mM) is needs as a control.
3. In the last line of page 8, the authors mentioned that they studied whether glycogen accumulation affected autophagy. This sentence is not consistent with the authors' conclusion that hyperglycemia impaired autophagy in Fig. 8.
4. The mechanism of decreased autophagy in islet cells with glycogen accumulation was not studied except downregulated beclin-1 expression. Is it possible to study at least one more autophagy regulator among AMPK, mTOR, PI3P accumulation, etc?

Points suggested by the Editor:

- A. Key results: Glycogen accumulates in islet cells with activating Katp channel, which leads to islet cell dysfunction and death.
- B. Originality: Original model of othe authors were employed.
- C. Data: Improved by the incorporation of the reviewers' comments.
- D. Statistics: no specific problem.
- E. Conclusion: valid and reliable.
- F. Suggested improvement: Improved by the incorporation of the reviewers' comments.
- G. References: no specific problem.
- H. Clarity and context: Improved by the incorporation of the reviewers' comments.

Reviewer #2 (Remarks to the Author):

The authors have addressed all issues raised in my review and this reviewer has no further comments.

Reviewer #3 (Remarks to the Author):

none

Reply to the Reviewers: referee 1

1. In page 9, autophagic bodies were mentioned. There is no clear definition of autophagic bodies or mention about how to count autophagic bodies. Are 'autophagic bodies' same as autophagosomes?

Yes. For clarity, we have now changed 'autophagic bodies' to 'autophagosomes'. We have also specified how autophagosomes were counted in the Methods section. "Autophagosomes were defined as double-membrane vesicles containing electron-dense membranes and other cytoplasmic material (Fig.5H). We also added the following to the Figure legend "The number of autophagosomes were counted from 10 fields of view per islet and expressed per μm^2 of beta-cell cytoplasm."

2. In Fig. 7J, Pcc1r3c expression in the presence of normal glucose (rather than 25 mM) is needs as a control.

Data for 5mM glucose have been added.

3. In the last line of page 8, the authors mentioned that they studied whether glycogen accumulation affected autophagy. This sentence is not consistent with the authors' conclusion that hyperglycemia impaired autophagy in Fig. 8.

We have modified the text on page 8. We now say, "We therefore next explored whether autophagy is impaired in diabetic βV59M islets."

4. The mechanism of decreased autophagy in islet cells with glycogen accumulation was not studied except downregulated beclin-1 expression. Is it possible to study at least one more autophagy regulator among AMPK, mTOR, PI3P accumulation, etc?

We agree with the reviewer that it would be interesting to determine the mechanism of decreased autophagy. However, this would require a considerable amount of time, as new mice would need to be bred, and is beyond the scope of this paper, whose main focus is glycogen. We therefore feel that this study would be more appropriate for a further paper focused specifically on autophagy.

Referees 2 and 3 had no further requests or comments.